# Crystal structure of steroid reductase SRD5A reveals conserved steroid reduction mechanism

Yufei Han [1,9], Qian Zhuang[2,9], Bo Sun[3,9], Wenping Lv [4,9], Sheng Wang[5,9], Qingjie Xiao[6], Bin Pang[1], Youli Zhou[1], Fuxing Wang[1], Pengliang Chi[6], Qisheng Wang[3], Zhen Li[7], Lizhe Zhu [4], Fuping Li[8], Dong Deng[6✉], Ying-Chih Chiang [1✉], Zhenfei Li [2✉] & Ruobing Ren [1✉]

Steroid hormones are essential in stress response, immune system regulation, and reproduction in mammals. Steroids with 3-oxo-$\Delta^4$ structure, such as testosterone or progesterone, are catalyzed by steroid 5α-reductases (SRD5As) to generate their corresponding 3-oxo-5α steroids, which are essential for multiple physiological and pathological processes. SRD5A2 is already a target of clinically relevant drugs. However, the detailed mechanism of SRD5A-mediated reduction remains elusive. Here we report the crystal structure of PbSRD5A from *Proteobacteria bacterium*, a homolog of both SRD5A1 and SRD5A2, in complex with the cofactor NADPH at 2.0 Å resolution. PbSRD5A exists as a monomer comprised of seven transmembrane segments (TMs). The TM1-4 enclose a hydrophobic substrate binding cavity, whereas TM5-7 coordinate cofactor NADPH through extensive hydrogen bonds network. Homology-based structural models of HsSRD5A1 and -2, together with biochemical characterization, define the substrate binding pocket of SRD5As, explain the properties of disease-related mutants and provide an important framework for further understanding of the mechanism of NADPH mediated steroids 3-oxo-$\Delta^4$ reduction. Based on these analyses, the design of therapeutic molecules targeting SRD5As with improved specificity and therapeutic efficacy would be possible.

[1] Kobilka Institute of Innovative Drug Discovery, School of Life and Health Sciences, The Chinese University of Hong Kong, Shenzhen, Guangdong 518172, China. [2] State Key Laboratory of Cell Biology, Shanghai Institute of Biochemistry and Cell Biology, Center for Excellence in Molecular Cell Science, Chinese Academy of Sciences, University of Chinese Academy of Sciences, 320 Yueyang Road, Shanghai 200031, China. [3] Shanghai Synchrotron Radiation Facility, Shanghai Advanced Research Institute, Chinese Academy of Sciences, Shanghai 201204, China. [4] Warshel Institute for Computational Biology, School of Life and Health Sciences, The Chinese University of Hong Kong, Shenzhen, Guangdong 518172, China. [5] Tencent AI lab, Shenzhen, Guangdong 518000, China. [6] Department of Obstetrics, Key Laboratory of Birth Defects and Related Disease of Women and Children of MOE, State Key Laboratory of Biotherapy, West China Second Hospital, Sichuan University, Chengdu 610041, China. [7] School of Science and Engineering, The Chinese University of Hong Kong, Shenzhen, Guangdong 518172, China. [8] Human Sperm Bank, Key Laboratory of Birth Defects and Related Disease of Women and Children of MOE, West China Second Hospital, Sichuan University, Chengdu 610041, China. [9] These authors contributed equally: Yufei Han, Qian Zhuang, Bo Sun, Wenping Lv, Sheng Wang. ✉email: dengd@scu.edu.cn; chiangyc@cuhk.edu.cn; zhenfei.li@sibcb.ac.cn; renruobing@cuhk.edu.cn

Steroid hormones, derived from cholesterol through de novo steroidogenesis in the adrenal cortex, the gonads, and the placenta[1,2], are essential in stress response, immune system regulation, and reproduction in mammals[3–5]. Steroids with 3-oxo-$\Delta^4$ structure could be converted to their corresponding 3-oxo-5$\alpha$ steroids by SRD5As for proper physiological function[6,7]. Abnormal activities of SRD5As will lead to benign prostatic hyperplasia, alopecia, prostatic cancer, or infertility owing to the poor quality of sperms. 4-androstene-3,17-dione and testosterone are catalyzed to 5$\alpha$-androstanedione and dihydrotestosterone (DHT), respectively, to regulate the development of prostate cancer[8–11]. Progesterone was catalyzed to 5$\alpha$-dihydroprogesterone for the generation of allopregnanolone, a neurosteroid to bind to GABA receptor[12,13]. The generation of 5$\alpha$-tetrahydrocortisol is involved in the inactivation and excretion of cortisol. It has been reported recently that SRD5As also participates into the metabolism of cytochrome P450 17A1 inhibitors used to treat castrate resistant prostate cancer, for example, abiraterone and galeterone[14,15].

Human steroid 5$\alpha$-reductase isozymes, belonging to NADPH-dependent oxidoreductase family, have three members (SRD5A1-3). SRD5A1 and -2 are mainly involved in the metabolism of steroid hormones[16], whereas SRD5A3 converts polyprenol into dolichol for the early steps of protein N-linked glycosylation[17,18]. Both SRD5A1 and -2 are endoplasmic reticulum membrane-embedded and utilize NADPH as a hydride (H⁻) donor for the $\Delta^4$ reduction reactions[19–21]. However, SRD5A1 prefers 4-androstene-3,17-dione as substrate to generate 5$\alpha$-androstanedione and SRD5A2 prefers testosterone to produce DHT[22,23]. During prostate cancer progression[24–26], decreased SRD5A2 activity and increased SRD5A1 activity have been observed and the related clinical significance remains to be explored[27]. Several inhibitors have been developed to inhibit the activity of SRD5A1 and -2 for disease treatment (Fig. 1a)[28,29]. Finasteride specifically inhibits SRD5A2 and dutasteride inhibits both SRD5A1 and -2[30], both of which are widely used for benign prostatic hyperplasia (BPH)[31–35]. Although SRD5A1 and -2 have been identified and extensively explored for decades, the structural information and molecular reaction mechanism are still poorly understood.

In this work, we report the crystal structure of PbSRD5A from *Proteobacteria bacterium* and homology-based structural models of human HsSRD5A1 and -2. Together with biochemical characterization of PbSRD5A and HsSRD5A2, our findings suggest the catalytic mechanism evolutionarily conserved in steroid 5$\alpha$-reductases catalyzing NADPH-mediated steroids reduction.

## Results

**Functional characterizations of PbSRD5A.** Some difficulties of structural study of human SRD5A1 and -2 were encountered owing to the low expression level and instability during protein expression and purification. To unveil the molecular mechanisms underlying the substrate recognition and catalytic reaction of steroid 5$\alpha$-reductases, BLAST searches[36], using human SRD5A1 and -2 (HsSRD5A1, and -2) as queries, against the sequenced bacterial genomes were performed to identify a steroid 5$\alpha$-reductase suitable for structural investigation. Elaborative selection was performed after the identification of hundreds of SRD5A candidates and four bacterial homologs, including the homolog from *P. bacterium* (PbSRD5A), were cloned, expressed, and purified using the insect cell expression system. Functions of all candidates were carefully examined and crystallization trials were conducted. PbSRD5A presented monodisperse behavior on size-exclusion chromatography and was selected for the structural and functional characterization due to its higher similarity to

HsSRD5As, highest expression level and best crystal quality (Supplementary Fig. 1a). PbSRD5A shared 60.6% and 51.5% sequence similarities with HsSRD5A1 and -2, respectively. To validate the function of PbSRD5A, steroid reduction assays were carried out. PbSRD5A presented potent activity in keto-steroid reduction, with obvious selectivity to progesterone, limited activity to 4-androstene-3,17-dione, but no detectable activity to testosterone, albeit with the similar structures among three steroids (Fig. 1b, c, Supplementary Fig. 1b and Table 1). Interestingly, dutasteride inhibited the reduction reaction of progesterone with a half-maximal inhibitory concentration ($IC_{50}$) of $1.59 \pm 0.19\ \mu M$ (Fig. 1d, e). However, finasteride was a weak inhibitor (Fig. 1d).

**Overall architecture of PbSRD5A.** After extensive crystallization trails, the crystals of PbSRD5A in space group $C222_1$ yielded in several conditions under monoolein lipid using lipidic cubic phase crystallization method (Supplementary Fig. 1c) and diffracted x-ray to 2.0 Å resolution. The structural model of PbSRD5A, derived from the modified framework of trRosetta[37] aided by a fused deep residual network that took input multiple sequence alignments, were applied as the initial phase for structure determination (Supplementary Fig. 1d). The structure was determined by molecular replacement using the predicted structural model (modelarchive ID: ma-xs1jw) and refined to 2.0 Å resolution (Fig. 2a, Supplementary Fig. 1e and Table 2).

PbSRD5A comprises seven transmembrane segments (TMs), with the N-termini located on the periplasmic side of the plasma membrane and C-termini on the cytosolic side. Given the level of sequence conservation, it is likely that all SRD5As exhibit the same fold as PbSRD5A (Supplementary Fig. 1f). We applied Dali search (http://ekhidna2.biocenter.helsinki.fi/dali/) using PbSRD5A structure and found that the TM2-7 of PbSRD5A can be superimposed with TM5-10 of the integral membrane delta (14)-sterol reductase MaSR1 (Protein Data Bank code: 4QUV) (Supplementary Fig. 2a), but TM2-4 of MaSR1 were missing in PbSRD5A (Supplementary Fig. 2b). Notably, the TM1 conformations of these two proteins were largely different (Supplementary Fig. 2b). The connecting sequence between TM1 and TM2 adapted a short $\beta$ strand ($\beta1–2$), which formed antiparallel $\beta$-sheets with the corresponding strand $\beta3-4$ linking TM3 and TM4 (Fig. 2a). The linkages between TM4, TM5, and TM6 adapted to short helices and stabilized the protein conformation through hydrogen bonds and hydrophobic interactions with other loops as well as C-termini (Fig. 2a).

There is one PbSRD5A molecule in each asymmetric unit. Examination of the crystal lattice revealed three interfaces, two of which are owing to the crystal packing to the neighboring symmetry-related molecules and the third interface is indeterminate (Supplementary Fig. 3a, b). This interface was mediated exclusively through van der Waals interactions between TM1 of two adjacent protomers (Supplementary Fig. 3c). To examine the oligomerization state of PbSRD5A in solution, we applied the PbSRD5A and a mycobacterial homolog of Insig protein (MvINS) (PDB code: 4XU4) to analytical ultracentrifugation (AUC) analysis. The theoretical molecular weight of PbSRD5A and MvINS protomers are 29.0 kDa and 23.6 kDa, respectively. The AUC results indicated that PbSRD5A functioned as a monomer (27.5 kDa) in solution and MvINS exhibited as a trimer (66.6 kDa) (Supplementary Fig. 3d), consistent with the previous cross-linking results[38].

**Coordination of NADPH.** With most amino acids assigned in the electron density map, two omitted electron densities were clearly visible in the central cavity near the cytoplasmic side (Fig. 2b). NADPH perfectly fitted into one of the densities mainly

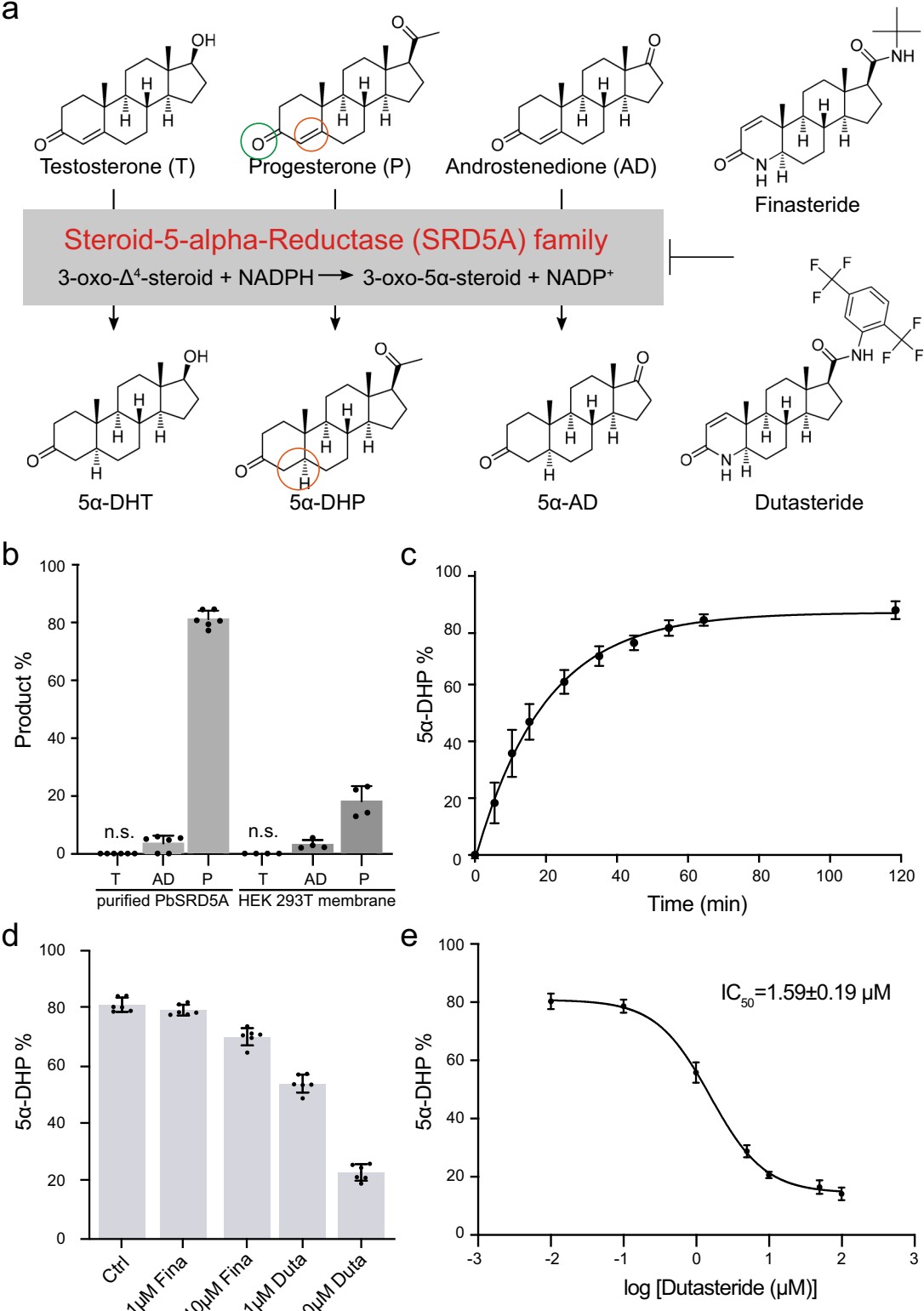

**Fig. 1 Steroid 5α-reductase activity of PbSRD5A. a** Steroid 5α-reductase family proteins can reduce $\Delta^4$ carbon–carbon double bound to form α configuration at $C^5$ of products. These reactions can be inhibited by finasteride and dutasteride specifically. **b** Testosterone (T), 4-androstene-3,17-dione (AD), and progesterone (P) metabolism in vitro and in HEK 293 T cell. Purified PbSRD5A and 293 T cells transiently transfected with PbSRD5A were treated with [$^3$H]-labeled T, AD, and P for 1 h and 12 h, respectively. n.s. means no signal. **c** Time curve of reduction reaction from progesterone to 5α-DHP by PbSRD5A. **d** Inhibition of PbSRD5A by finasteride (Fina) and dutasteride (Duta) in vitro, both with the concentration of 1 μM and 10 μM, respectively. **e** Dutasteride inhibits the activity of PbSRD5A in vitro, with an IC50 value of $1.59 \pm 0.19$ μM (logIC50 = $0.20 \pm 0.05$). **b–e** The percentages of products were measured using high-performance liquid chromatography (HPLC) with detected radioactivity. Data are mean±s.d. derived from technically independent experiments in duplicate. Each experiment was reproduced three times on separate occasions with similar results.

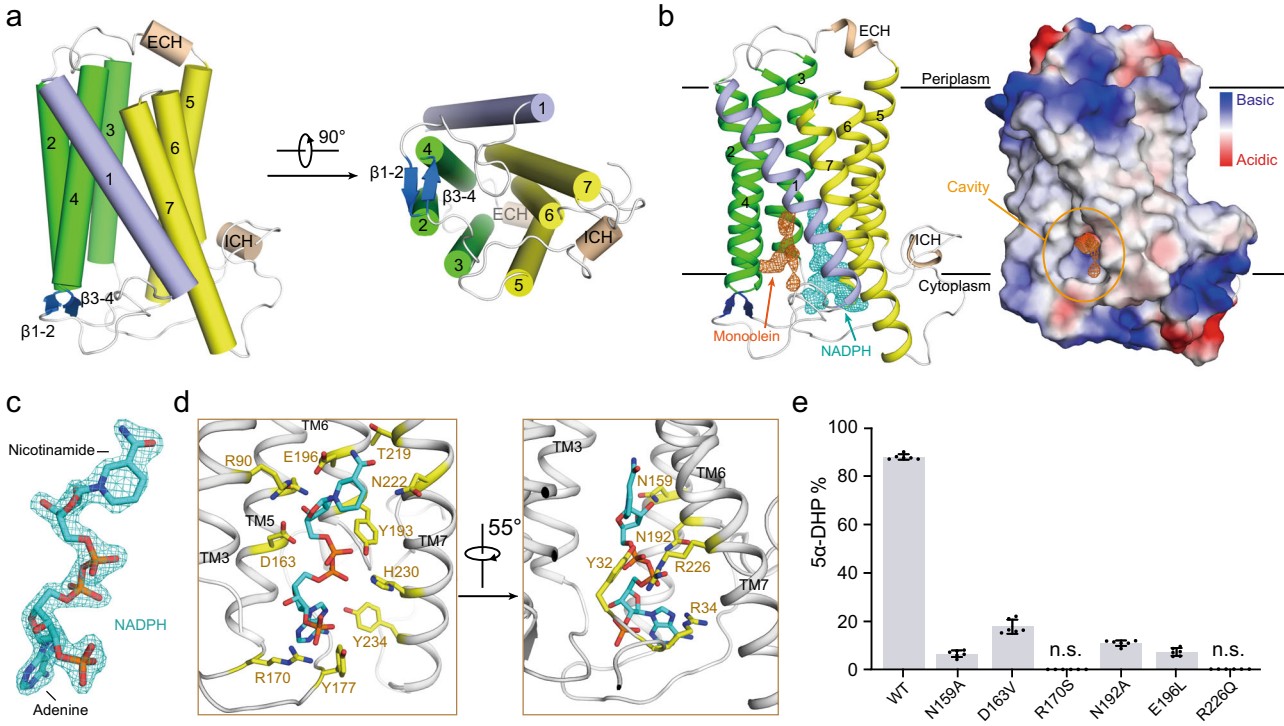

**Fig. 2 The crystal structure of PbSRD5A and coordination of NADPH. a** The overall structure of PbSRD5A. Two perpendicular views are shown. The seven transmembrane segments are divided into TM1 (lightblue), TM2-4 (green), and TM5-7 (yellow). The extracellular and intracellular short alpha helices (ECH and ICH) are colored wheat. The short antiparallel beta strands are colored blue. The loop regions are colored white. **b** 2Fo-Fc map for monoolein (orange mesh) and NADPH (cyan mesh) are both contoured at 1.0σ. The two black lines show the approximate location of the lipid bilayer. The cavity is circled in an electrostatic surface representation. **c** NADPH fits into the density map shown in **b**. Nicotinamide and adenine groups are labeled. **d** Coordination of NADPH by polar residues of PbSRD5A. The residues that are hydrogen-bonded to NADPH are shown in yellow sticks. **e** Functional validation of residues coordinating NADPH in PbSRD5A. Enzyme activities were measured by the percentage of 5α-DHP reduced from [3H]-labeled progesterone in vitro within 1 h detected by HPLC. Data are mean±s.d. derived from technically independent experiments in duplicate. Each experiment was reproduced three times on separate occasions with similar results.

coordinated by TM5-7 of PbSRD5A (Fig. 2b, c). NADPH formed extensive hydrogen bonds with N159, D163, and R170 on TM5, N192, Y193, and E196 on TM6, and T219, N222, R226, and H230 on TM7 (Fig. 2d and Supplementary Fig. 4a, b). The residues on intracellular loop 1 (Y32 and R34) and loop 3 (Y177) also contributed to NADPH binding through direct hydrogen bonds (Fig. 2d and Supplementary Fig. 4a, b). Some residues, such as R34 on intracellular loop 1, V100 on intracellular loop 2, N192 on TM6 and R226 on TM7 partly interacted with NADPH through water mediated hydrogen bonds (Supplementary Fig. 4c). In addition, the residues W50 on TM2, F94 and M98 on TM3, L166 and L169 on TM5, and L223 on TM7 interacted with NADPH by hydrophobic effects (Supplementary Fig. 4c). The results of in vitro reduction assay demonstrated that mutations in these residues of PbSRD5A impaired the conversion from progesterone to 5α-dihydroprogesterone, indicating the crucial role of these residues for reductase activity (Fig. 2e). The other elongated strip density was modeled with the crystallization lipid monoolein, whose alkenyl "tail" is accommodated well (Fig. 2b and Supplementary Fig. 5a) in the hydrophobic pocket majorly composed by the residues located on TM1-4 (Supplementary Fig. 5b). The residue Q53 formed the hydrogen bond with the glycerol "head" of monoolein (Supplementary Fig. 5b). Considering of the hydrophobicity of steroid substrates and the orientation of NADPH, more specifically the position of nicotinamide, we hypothesized that monoolein probably occupied the steroid substrates binding pocket. A topology model of PbSRD5A is drawn to better represent the structural organization (Supplementary Fig. 1g).

**Molecular mechanism of 3-oxo-Δ4 reduction.** Considering the sequence similarities of bacterial SRD5A, HsSRD5A1, and -2, we built the homology models of HsSRD5A1 (modelarchive ID: ma-bfecj) and -2 (modelarchive ID: ma-ib3wq) using the server for initial model building of PbSRD5A (Fig. 3a, b and Supplementary Fig. 6). The highly conserved residues were mainly concentrated to the NADPH and the putative steroid-binding pockets, as well as the interfaces of TMs (Supplementary Fig. 6). It suggested the evolutionary conservation of NADPH binding and substrate recognition other than the overall architectures. The movements of TM1s were observed after we superimposed TM2-7 of PbSRD5A structure with HsSRD5A1 and -2 models (Supplementary Fig. 7a).

To fully explore the substrate recognition and reduction reaction mechanism, progesterone was docked into PbSRD5A, as well as testosterone into HsSRD5A1 and -2, respectively. For instance, in the SRD5A2 docking model, the native substrate testosterone can be accommodated in the semi-closed pocket composed by SBD (TM1-4), which is similar to that in PbSRD5A occupied by monoolein (Figs. 2b and 3c). This binding pose was also observed in the catalytic pocket of soluble steroid 5β-reductase3 (AKR1D1, Protein Data Bank code: 3COT), bound with progesterone (Supplementary Fig. 7b)[39]. Notably, the steroid-binding pockets of both enzymes contain a signature motif forming triangular hydrogen bonds that coordinate the ketone group at C-3 in the steroids (Fig. 3d, e). This coordination brings the substrate into close proximity to the nicotinamide of NADPH, leading to the hydride (H−) transfer and Δ4 double-bond reduction (Fig. 3d, e). For HsSRD5A2, this signature motif

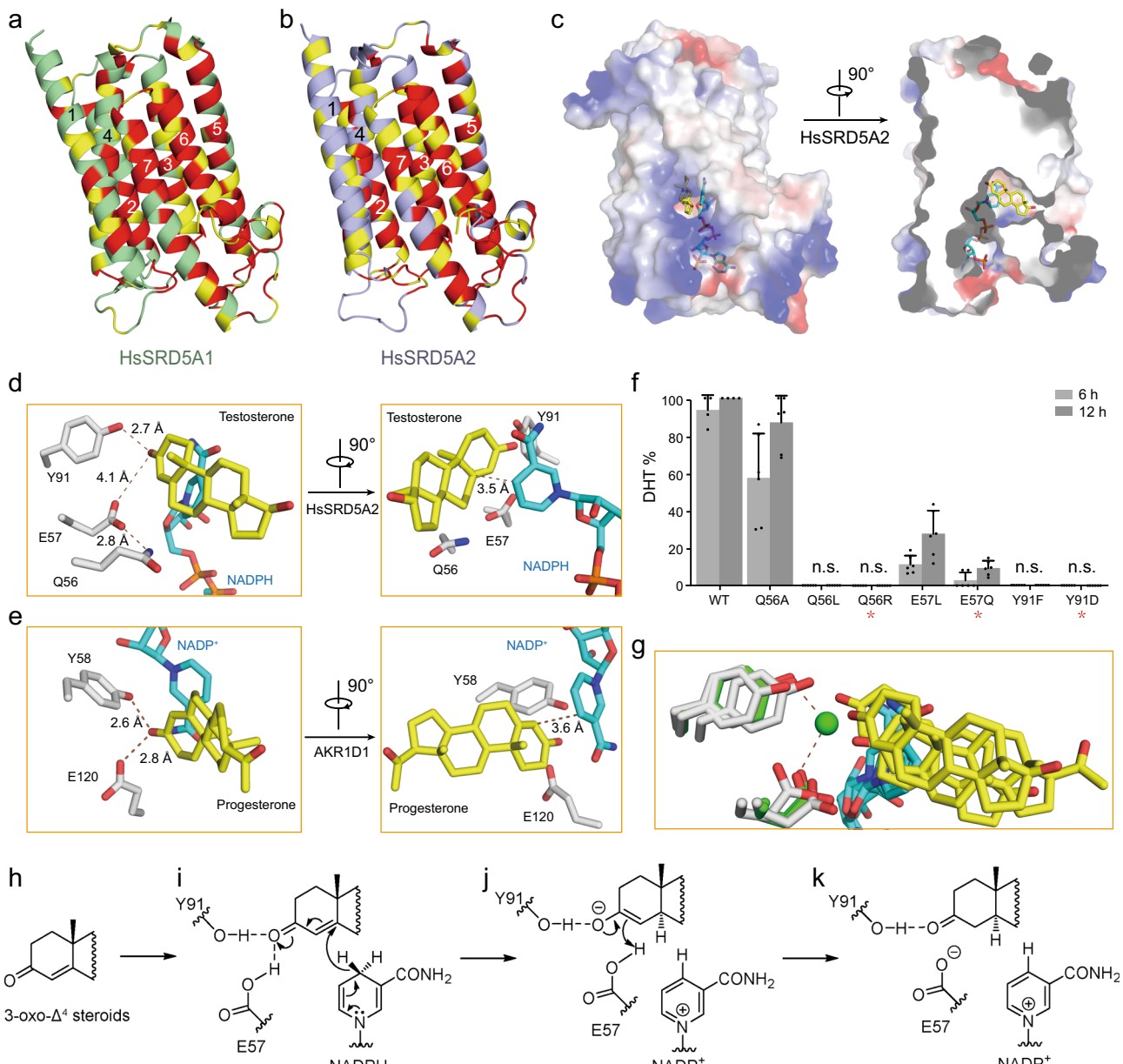

**Fig. 3 HsSRD5A1 and -2 modeling, substrate docking, and functional characterization. a** HsSRD5A1 homology model was generated on the basis of PbSRD5A structure and represented in palegreen. Invariant and conserved residues are colored in red and yellow, respectively. **b** HsSRD5A2 homology model was generated on the basis of PbSRD5A structure and represented in lightblue. **c** The docking pose of testosterone in HsSRD5A2 structural model. The semi-transparent electrostatic surface of HsSRD5A2 is shown. Testosterone is shown as yellow stick. NADPH is colored cyan. **d** The coordination of conserved Q-E-Y motif with testosterone in HsSRD5A2 docking model. Two perpendicular views are shown. **e** The coordination of conserved Q-E-Y motif in AKR1D1-progesterone complex structure. **f** Biochemical characterization of the Q-E-Y motif of HsSRD5A2. All variants were transient expressed in HEK293T cells for 24 h. The cells were treated with [3H]-labeled T for 6 or 12 h. The collected and homogenized cell membranes were treated with [3H]-labeled T. HsSRD5A2 activities were measured by the percentage of DHT detected by HPLC. Data are mean ± s.d. derived from technically independent experiments in duplicate. Each experiment was reproduced three times on separate occasions with similar results. Red stars (*) indicate the variants in patients. **g** The water molecule, coordinated by the Q-E-Y motif in PbSRD5A structure, was shown in green sphere. The water molecule occupied the space for the C-3 carbonyl oxygen in the docking results shown in Fig. 3d and Supplementary Fig. 7d, f. **h** Key structural features of 3-oxo-Δ⁴ steroids. **i** The tyrosine (Y91 in HsSRD5A2) and glutamate (E57 in HsSRD5A2) may form hydrogen bonds to the carbonyl oxygen of substrates. Y91, as a super acidic hydrogen bond donor, helps to activate α, β-unsaturated ketone moiety of substrates. **j** The proposed mechanism of reduction involves a two-step reduction by the enzyme and NADPH. In the first step, NADPH coordinates on α face of the substrate and adds a hydride to C5, leading to a selective reduction at C5 to form an enolate ion intermediate. **k** Second, the resonance-stabilized enolate ion is protonated by E57 at C4 and thus releasing a saturated ketone product and an NADP⁺.

includes Y91 and E57 (Fig. 3d) and for AKR1D1, Y58 and E120 (Fig. 3e)[39]. Additionally, Q56 of HsSRD5A2 bonded with E57 to stabilize the local conformation in HsSRD5A2 (Fig. 3d). In the reductase activity assay using cells containing HsSRD5A2

mutants, Q56A/L, E57L, and Y91F lost reductase activity unambiguously (Fig. 3f) with equally expression level (Supplementary Fig. 8), indicating the importance of these residues in substrate recognition and reduction. The Q56R, E57Q, and Y91D

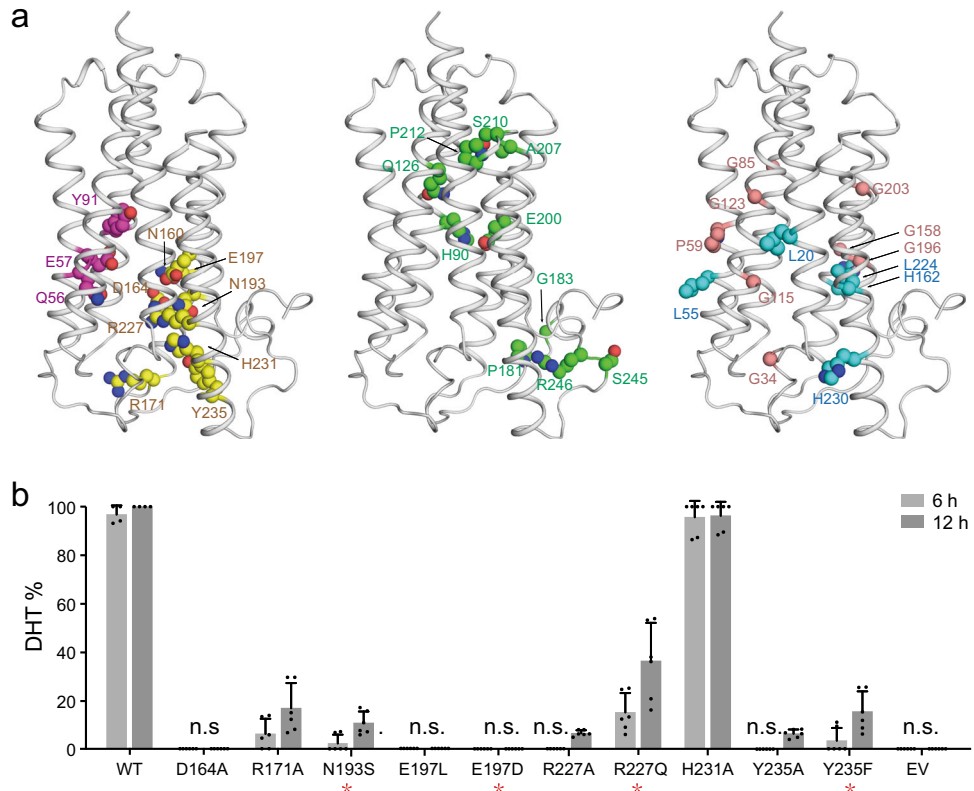

**Fig. 4 Disease-related loss-of-function mutations in HsSRD5A2 model. a** 32 disease-related loss-of-function mutations were classified into five categories. Catalytic site residues are in magenta, NADPH-binding residues are in yellow, structural destabilizing mutations are in green, wheat, or cyan based on residue properties. **b** Biochemical analysis of disease-related HsSRD5A2 variants. All variants were transient expressed in HEK293T cells for 24 h. The cells were treated with [3H]-labeled T for 6 or 12 h. The collected and homogenized cell membranes were treated with [3H]-labeled T. HsSRD5A2 activities were measured by the percentage of DHT detected by HPLC. Data are mean±s.d. derived from technically independent experiments in duplicate. Each experiment was reproduced three times on separate occasions with similar results. Red stars (*) below variant labels indicate the variants in patients.

variants, whose activities were also largely aborted in the reductase assay (Fig. 3f), were found in SRD5A2 deficient disease patients, such as steroid 5α-reductase 2 deficiency[40–43]. The "QEY" signature motif is highly conserved in all 5α-reductases we studied, including PbSRD5A and HsSRD5A1, demonstrating the evolutionarily conserved steroid reduction mechanism (Supplementary Fig. 7c–f). In the substrate-free PbSRD5A structure, Y87 donates a hydrogen bond to a water molecule, which in turn hydrogen bonded with E54. Interestingly, this hydrogen-bonded water molecule occupies a position close to that of carbonyl oxygen of substrates observed in all the docking models, which is consistent with the observation in AKR1D1 structures (Fig. 3g)[39].

**Disease-related mutagenesis of HsSRD5A2.** Owing to the essential function of SRD5A2 in steroidogenesis, large numbers of disease-related loss-of-function mutations have been identified and confirmed in vivo[44,45]. Guided by a homologous structural model of HsSRD5A2, we classified 32 mutations into five categories (Supplementary Table 3) and mapped on the HsSRD5A2 structural model (Fig. 4a). As discussed previously, the residues Q56, E57 and Y91 colored in magenta are located in the catalytic pocket and essential for the substrate binding and catalytic activity (Fig. 4a left). The residues for NADPH binding, including N160, D164, R171, N193, E197, R227, H231, and Y235, are colored in yellow (Fig. 4a left). The corresponding residues in PbSRD5A examined by in vitro reduction assay are all identical to HsSRD5A2 (Fig. 2e and Supplementary Table 3). The residues which form extensive hydrogen bond network to stabilize the protein conformation, but not directly interact with substrates and NADPH, are also highlighted (Fig. 4a

middle). Notably, the variants P181L, S245Y, and R246Q/W in the C-termini are also disease-related, indicating the C-terminal local conformation is crucial for reductase activity. Considering these residues are close to the NADPH binding site, these variants may destabilize the NADPH binding (Fig. 4a middle). Besides, numbers of Gly to X (colored in wheat) and X to Gly (colored in cyan) variants are found in steroid 5α-reductase 2 deficiency patients (Fig. 4a right). These mutations are mainly located on the TM interfaces and may reduce the HsSRD5A2 activity by disrupting the tertiary structure. The disease-related mutants are also extensively studied by steroid reduction assay (Fig. 4b). Except for H231A, the activities of all other variants decreased significantly. The structural analysis combining with biochemical studies clearly elucidated how disease-related mutations abort HsSRD5A reductase activity at the molecular level.

## Discussion

Enzymes of steroid 5α-reductase family, which catalyze the 3-oxo-$\Delta^4$ reduction reactions, have critical roles in steroidogenesis. Notably, all SRD5A enzymes share high sequence similarity from bacteria to mammal (Supplementary Fig. 9). On the basis of our structural observations and biochemical analysis, we firmly believed that PbSRD5A is a proper prototype to propose the catalytic model to shed light on the mechanistic understanding of all SRD5A enzymes.

It is not surprising that SRD5As exhibit similar fold and the NADPH cofactor-binding sites are highly conserved in the steroid reductase families. The structural and biochemical characterizations reported herein illuminated features of steroid

substrate recognition, the catalytic mechanism, and positions of natural mutations associated with steroidogenesis deficiency. Our results suggested that, if the C-3 carbonyl oxygen atom of the substrate displace water molecules in SRD5A substrate-binding pocket, the side chain of conserved tyrosine (Y91 in HsSRD5A2), together with adjacent protonated glutamate (E57 in HsSRD5A2), help to enolize α, β-unsaturated ketone moiety of substrates for hydride transfer from NADPH to C5 (Fig. 3h–j). Y91F/D and E57L mutants lost reductase activities unambiguously (Fig. 3f). This mechanism is also conserved in testosterone reduction reaction of AKR1D1[39,46]. Except for the hydride (H⁻) provided by NADPH, one additional proton is necessary to be transferred to C4 of testosterone to complete the $\Delta^4$ carbon–carbon double-bond reduction. Based on the structure model of HsSRD5A2, an oxygen atom (~15 Å³) is used as a probe to search the free space volume around E57 through the molecular dynamics (MD) simulation of HsSRDA2 (Supplementary Figure 10). In the presence of substrate, the extra space (<15 Å³ in most frames) is limited to accommodate additional water molecule (~30 Å³) in the catalytic site to provide proton. In addition, the biochemical analysis of E57Q mutant, which lost the deprotonation property of side chain, showed only 5% activity compared with wild type (Fig. 3f). These results indicated that the proton may be transferred from the protonated glutamate (Fig. 3k).

In terms of future research, one major question awaits further investigation: the substrate and inhibitor selectivity of SRD5As. As mentioned above, PbSRD5A mainly responded to progesterone, whereas HsSRD5A1 and -2 had significant reduction activity to 4-androstene-3,17-dione and testosterone, respectively (Fig. 1b, c). Moreover, dutasteride showed higher inhibitory potency than finasteride, which is similar to SRD5A1 but not -2 (Fig. 1d)[28]. We noticed that the cytosolic half of TM1 and -2 of all SRD5As may play critical roles in recognizing the "tail" part attached to C-17 of steroid substrates. Intriguingly, the movements of TM1s (Supplementary Fig. 7a), combined with the observation of significantly large B factor value of TM1 in PbSRD5A structure (Supplementary Fig. 1f), suggested that the TM1 may play important roles in the substrate entering and releasing. Based on the sequence alignment over 150 SRD5As, the cytosolic half of TM1 and -4 are the most variable regions (Supplementary Fig. 6 and 9). The sequence diversity on TM1 and -4 may also contain the recognition "code" contributing to the specificity for varies of steroid substrates, as well as varies of inhibitors. Although our knowledge of these enzymes is not yet complete, the results we described here should serve as a foundation for future exploring the substrate and inhibitor selectivity, which is one of the determinants to successfully design therapeutic molecules targeting SRD5As with improved specificity and therapeutic effects.

## Methods

**Cloning and expression of PbSRD5A.** The cDNA of *P. bacterium* SRD5A (GenBank ID RIL07334.1, denoted as *PbSRD5A*) was synthesized from Genescript and subcloned into pFastBac vector (Invitrogen) with amino-terminal 10XHis tag. Wild type and all PbSRD5A mutants were generated with a standard PCR-based strategy. Primers used in cloning are listed in Supplementary Table 4. Baculovirus was generated with the Bac-to-bac system (Invitrogen) and used for infecting *Spodoptera frugiperda* (Sf9) cells (Sino-Bio) at density of $2 \times 10^6$ cells per mL and 10 mL of virus per liter of cells. Infected cells were collected after 60 h by centrifugation, frozen in liquid nitrogen and stored at −80 °C.

**Purification of PbSRD5A.** Cell pellets were disrupted using Dounce tissue grinder (homogenizer) for 60 cycles on ice with lysis buffer (25 mM HEPES pH 7.5, 150 mM NaCl, 5% v/v glycerol) and subjected to ultracentrifugation at $150,000 \times g$ for 60 min at 4 °C. The supernatant was discarded and the membrane fractions were homogenized in 20 mL solubilization buffer (25 mM HEPES pH 7.5, 150 mM NaCl, 5% v/v glycerol) with cocktail inhibitors (Sigma) per initial liter of culture volume. Membrane fractions was dissolved by the addition of 2% (w/v) n-dodecyl-β-D-maltopyranoside (DDM, Bluepus) and rotated at 4 °C for 2 h. Cell debris was pelleted at $40,000 \times g$ for 30 min at 4 °C, and the supernatant was loaded on

Ni²⁺-nitrilotriacetate affinity resin (Ni-NTA, Qiagen), 0.5 mL resin per liter cell culture. The resin was washed twice with 40 column volumes of wash buffer (25 mM HEPES pH 7.5, 150 mM NaCl, 5% v/v glycerol, 30 mM imidazole pH 8.0, 0.02% w/v DDM), and eluted with elution buffer (25 mM HEPES pH 7.5, 150 mM NaCl, 5% v/v glycerol, 250 mM imidazole pH 8.0, 0.02% w/v DDM). After concentrating, the protein was treated by 0.5 mg/mL drICE protease to remove the 10xHis tag and incubated with 2 mM NADPH (Sigma-Aldrich) for 2 h at 4 °C. The protein was applied to size-exclusion chromatography on a Superose 6 Increase 10/300 GL column (GE healthcare) in gel filtration buffer (25 mM HEPES pH 7.5, 150 mM NaCl, 5% v/v glycerol, 0.02% w/v DDM, 5 mM DTT). Fractions containing the highest concentration of PbSRD5A were collected for crystallization.

**Lipid cubic phase crystallization.** Purified PbSRD5A protein was concentrated to 25 mg/mL and incubated with another 10 mM NADPH before preparing the cubic phase. We reconstituted the protein into lipid cubic phase by mixing with molten monoolein at a protein/lipid ratio of 2:3 (w/w) using a syringe lipid mixer[47]. The meso-phase was dispensed in 60 nL drops onto 96-well glass plates and overlaid with 800 nL precipitant solution. The crystallization trials were performed using a Gryphon robot arm (Art Robbins Instruments). Crystals grew from buffers containing 40% (v/v) PEG 400, 100 mM HEPES pH 7.5, 100 mM potassium acetate, and reach full size (15 × 15 × 120 μm) within one week at 20 °C.

**Model building of PbSRD5A, human SRD5A1, and -2.** The key components of 3D model construction pipeline consist of three parts: (i) multiple sequence alignment (MSA) generation from a diverse set of sequence databases under different sequence search engines[48], (ii) a fused deep residual network that takes multiple MSAs as input to predict inter-residue distances and orientations, and (iii) a fast Rosetta model building protocol based on restrained minimization with distance and orientation restraints[37]. Using PbSRD5A as an example, nine alternative MSAs were generated from three sequence databases: UniClust30[49], UniRef90[50], and NCBI_NR[51], and for each database we used HHblits[52], JackHMMER[53], and PSI-BLAST[54] to search for homology sequences. To fuse these nine alternative MSAs, we implemented a deep residual network with a strip pooling module to effectively capture long-range relationship of residual pairs. Such fuse architecture could alleviate the issue caused by a problematic MSA that could decrease the prediction accuracy. Following trRosetta[37] and AlphaFold[55], we generated the 3D structure model from the predicted distance and orientation using constrained minimization. Specifically, the predicted distance- and orientation- probabilities are first converted into potentials, which are then used as restraints to be fed into Rosetta together with centroid level (coarse-grained) energy optimization. Afterwards, the top 50 folded structures satisfying the restraints were selected according to Rosetta energy for further full-atom relaxation by Rosetta. Then, five best models were selected from the 50 structures using GOAP[56] energy. The best model were selected and submitted to QMEANDisCo[57] for quality evaluation.

This workflow was then applied to build human SRD5A1 and -2 models. Using PbSRD5A structural model as a template, the homology models of HsSRD5A1 and -2 were generated by RosettaCM[58]. Similarly, trRosetta was used to refine the 3D models from the predicted distance and orientation using constrained minimization. Five best models were selected from the 50 structures using GOAP energy and the top-ranked models were submitted to QMEANDisCo for quality evaluation. All structural models we used were deposited to Modelarchive.

**Data collection and structure determination.** The X-ray diffraction data set was collected at SSRF (Shanghai Synchrotron Radiation Facility) beamlines BL18U1. The MD2 micro-diffractometer (Arinax, Ltd.) and Pilatus3 X-6M detector (Dectris, Ltd.) were used to collect the data. The beam size is ~30 × 30 μm at the spot position with the flux about $8 \times 10^{11}$ phs/s. The data set was further integrated and scaled with XDS Package[59] and Aimless in the CCP4[60] suite, respectively. The structure of PbSRD5A was solved by molecular replacement (MR) with the predicted model as search model using PHASER[61]. The structure was refined by PHENIX[62] and Coot[63]. The related statistics are listed in Supplementary Table 2.

**In vitro enzymatic assay of PbSRD5A.** Purified wild-type or mutated PbSRD5A of 100 ng were incubated with [³H]-labeled progesterone (~500,000 cpm, 7.5 nM), progesterone (10 nM) and dutasteride (10 μM) or finasteride (10 μM) in 0.2 ml of PBS buffer with 0.02% DDM (pH 7.4) and 0.2 mM NADPH at 37 °C for 1 h. The reaction was stopped by the addition of 0.5 ml ethylacetate: isooctane (1:1). Steroids were extracted and analyzed by high-performance liquid chromatography (HPLC). PbSRD5A in vitro metabolism assay for other two substrates, testosterone and 4-androstene-3,17-dione, was performed using the same protocol. All experiments have been repeated for three times and the results represent mean and standard deviation (s.d.).

**HPLC analysis.** HPLC analysis was performed on a Waters Acquity ARC HPLC (Waters, Ireland). Dried samples were reconstituted in 100 μL of 50% methanol and injected into the HPLC. Metabolites were separated on CORTECS C18 reverse-phase column 4.6 × 50 mm, 2.7 μM (Waters, Ireland), using a methanol/water gradient at 40 °C. The column effluent was analyzed using β-RAM model 3 in-line radioactivity detector (LABLOGIC, USA). Results represented the mean and standard deviation (s.d.) value from three repeated experiments. All HPLC studies were run in duplicate and repeated at least three times independently.

**Enzymatic assay of HsSRD5A2**. HEK293T cells were purchased from the American Type Culture Collection (Manassas, VA) and maintained in Dulbecco's Modified Eagle Medium with 10% FBS (ExCell Bio, China). All experiments were done in plates coated with poly-DL-ornithine (Sigma-Aldrich, St. Louis, MO). Cell lines were authenticated by Hybribio (Guangzhou, China) and determined to be mycoplasma free with primers 5′-GGGAGCAAACAGGATTAGATACCCT-3′ and 5′-TGCACCATCTGTCACTCTGTTAACCTC-3′.

Cells were seeded and incubated in 24-well plate for 24 h before transfected with the indicated plasmids. Cells were then treated with a mixture of [$^3$H]-labeled steroids (final concentration, 10 nM T and 10 nM progesterone; ~1,000,000 cpm/well; PerkinElmer, Waltham, MA) at 37 °C. Aliquots of medium were collected and treated with 300 units of β-glucuronidase (Novoprotein Scientific Inc, China) at 37 °C for 2 h, extracted with ethylacetate:isooctane (1:1), and dried in a freeze dryer (Martin Christ Gefriertrocknungsanlagen, Germany). Steroids were extracted and analyzed by HPLC.

**Substrate docking and MD simulation for PbSRD5A, HsSRD5A1, and -2**. Optimized binding poses of the pair of ligands and modeled SRD5A1/2-NADPH complexes were predicted by iteratively employing molecular docking using Auto-Dock Vina[64] and MD simulations using GROMACS[65]. The NADPH was initially placed in the models by structural alignment of the HsSRD5A1/2 models and PbSRD5A structure; and the testosterone was docked into the initial structure of the binary complex using AutoDock Vina[64]. The ternary complex was embedded in a lipid bilayer constructed according to the compositions of ER membrane. The whole system was solvated in a TIP3P water with 0.15 M of NaCl, following by 200 ns of restrained MD simulations to fully relax and equilibrate the solvent and membrane structure at 303.15 K and 1 bar. Namely, a harmonic potential (500 kJ/mol) was applied to restraint the movement of the heavy atoms of the NADPH molecule, to avoid artificial/unphysical conformational deformation of the conserved binding pose of the NADPH inside proteins in our simulations; a harmonic potential (50 kJ/mol) was applied to constraint the distance of the center-of-mass of between the nicotinamide group of NADPH and the C $=$ O group of substrates predicted by the initial docking pose, to improve the efficiency of our MD sampling around the reaction "active state" of the system. Here, protonation state of the protein was assigned by the web server H++[66–68] assuming pH 7.4, and charmm36m force field[69] was employed in simulations. Finally, 50 ns of plain MD simulation without any restraint was carried out to capture the thermodynamics effect on the conformational fluctuation of the complex. The snapshots of the protein/NADPH complex were recorded every 100 ps in our production MD, generating 500 conformers in total for each complex. Then, molecular docking was carried out for each of the 500 conformers generated from our MD simulations using AutoDock Vina. The protonation states of the substrates and the NADPH-protein complexes were kept as in the MD simulations. A 25 × 25 × 25 Å$^3$ box centered at the ring of nicotinamide group of NADPH, was considered as the searching region for each conformer. Flexibilities of the substrates and all side chain of the residues within the searching region were considered. Exhaustively (80) docking was carried out to search the optimized binding pose, with the so-called Broyden-Fletcher-Goldfarb-Shanno algorithm for local optimization. For data analysis, all the docked conformers were sorted according to the binding affinities scored by Vina. The docked result with the strongest binding affinity among the 500 conformers was considered as the first representative binding pose. Clustering analysis was performed to group the 500 conformers, considering the coordinates of only the heavy atoms of proteins. The representative of a cluster of conformers, which had the highest binding affinity on average, was considered as the second representative binding pose. The correlation between predicted binding affinity and structural displacement of the docked ligand from the one in the first binding pose was assessed by calculating their root-mean-square distance for the ligand aligned according to the heavy atoms of the protein.

**AUC analysis**. Sedimentation velocity analysis was carried out with an XL-I analytical ultracentrifuge (Beckman Coulter) with An-50 Ti rotor for PbSRD5A and MvINS at 20 °C. Reaction buffer containing 25 mM HEPES pH 7.5, 150 mM NaCl, 5% v/v glycerol, 0.02% w/v DDM, and 5 mM DTT was used while the reference solution was without DDM. All data were collected at a speed of 186,000 × $g$. Samples were monitored real time using UV absorption at 280 nm and optical refraction at intervals of 4 min. Buffer along was used as reference. The molecular weights were calculated by the SEDFIT[70] and GUSSI programs.

**Reporting summary**. Further information on research design is available in the Nature Research Reporting Summary linked to this article.

## Data availability
Data supporting the findings of this manuscript are available from the corresponding author upon reasonable request. Structure factors and atomic coordinates of PbSRD5A protein have been deposited in the Protein Data Bank under accession code 7C83. Initial models of PbSRD5A and two human SRD5As have been deposited in *modelarchive* with model IDs ma-xs1jw (PbSRD5A), ma-bfecj (human SRD5A1), and ma-ib3wq (human SRD5A2). PDB entries (4QUV, 4XU4, 3COT) used in this study were downloaded from Protein Data Bank. Source data are provided with this paper.

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

## Acknowledgements

We are grateful to Shanghai Synchrotron Radiation Facility (SSRF) for providing x-ray beam time and onsite assistance. We also thank Guijuan Cheng and Jie Ni at the Chinese University of Hong Kong, Shenzhen for critical discussion, figure and manuscript revision. We thank Wenqi Li and Wendan Chu to provide technical support in AUC analysis. This work was supported by funds from the National Key R&D Program of China (2018YFA0508200 to Z. Li; 2016YFA0502700 to D. Deng) from the Ministry of Science and Technology, National Natural Science Foundation of China (Project 31971218 to R. Ren and 21505134 to W. Lv), the Strategic Priority Research Program of Chinese Academy of Sciences (XDB19000000 to Z. Li), and Science, Technology and Innovation Commission of Shenzhen Municipality (Projects JCYJ-20180307151618765 and JCYJ-20180508163206306 to R. Ren). R. Ren was also supported in part by Kobilka Institute of Innovative Drug Discovery and Presidential Fellowship at the Chinese University of Hong Kong, Shenzhen. F. Wang and Y. Zhou was supported by Ganghong Youth Scholarship in CUHKSZ.

## Author contributions

R.R, D.D. and F.L. initiated the project. F.W. and Y.Z. aligned and ordered all genes. Y.H. and P.C. made all PbSRD5A constructs and purified the proteins. Y.H. and B.P. crystallized the protein. B.S and Q.X. collected and analyzed all diffraction data, and determined the structure. Z.L. and S.W. provided the initial model. Y.C., W.L. and L.Z. performed docking, MD simulations based on structural and biochemical results, and energy minimization. Q.Z. performed all radio isotope HPLC analysis. R.R., Z.L. and D.D. supervised the project. R.R., D.D. and Z.L. prepared the manuscript.

## Competing interests

The authors declare no competing interests.
