## [Peer Review File · Nature Communications]

REVIEWER COMMENTS

Reviewer #1 (Remarks to the Author):

Steroid 5 α -reductases (SRD5as) are enzymes involved in catalyzing steroids with 3-oxo- Δ 4 structure, such as testosterone, androstenedione and progesterone, to generate corresponding 3-oxo-5 α steroids, which are essential for multiple physiological and pathological processes. Abnormal activities of SRD5as have been linked to several diseases, such as benign prostatic hyperplasia, prostatic cancer and male infertility. Recently, SRD5as have also been indicated as potential targets for the treatment of COVID-19. Although human steroid 5 α -reductase isozymes, HsSRD5A1 and HsSRD5A2, have been extensively investigated for decades, the structural information and molecular reaction mechanism remain poorly understood. In this paper, the authors reported the crystal structure of PbSRD5A in complex with the cofactor NADPH at high resolution (2.0 Å). PbSRD5A is a bacterial homolog of SRD5A from Proteobacteria, which shares 60.6% and 51.5% sequence similarities with HsSRD5A1 and HsSRD5A2, respectively. Homology-based structural models of HsSRD5A1 and HsSRD5A2 were built, and substrates were docked to PbSRD5A, HsSRD5A1 and HsSRD5A2, respectively. Extensive biochemical characterizations were also carried out, using in vitro reduction assay for PbSRD5A and cell-based enzymatic assay for HsSRD5A2. These structural, computational and biochemical studies were combined to understand the mechanism of NADPH mediated steroids 3-oxo- Δ 4 reduction. The presentation of the manuscript should be significantly improved to make the manuscript easier to follow. There are also many typos.

Major concerns:

1. On Page 4, in the section on "Functional characterizations of PbSRD5A", the authors stated that "To unveil the molecular mechanisms underlying the substrate recognition and catalytic reaction of SRD5A1 and -2, BLAST searches, using HsSRD5A1 and -2 as queries, against the sequenced bacterial genomes were performed to identify a steroid 5 α -reductase suitable for structural investigation." It seems that the authors aimed to unveil the mechanisms of substrate recognition and catalytic reaction of HsSRD5A1 and -2. However, they selected PbSRD5A, a bacterial homolog, as the target of investigation. Please explain in the main text why HsSRD5A1 or -2 were not selected as the target for investigation directly.

2. On Page 8, in the section on "Molecular mechanism of 3-oxo- Δ 4 reduction", the authors stated that "Considering the sequence similarities of bacterial SRD5A, HsSRD5A1 and -2, we built the homology models of HsSRD5A1 and -2 ..." and "To fully explore the substrate recognition and reduction reaction mechanism, progesterone was docked into PbSRD5A, as well as testosterone into HsSRD5A1 and -2, respectively." The substrate selectivity suggests that different SRD5As may adopt different conformations in the substrate binding sites. Indeed, sequence alignment analysis of SRD5as across different species indicated that while the residues involved in the binding of NADP are highly conserved, the residues in the progesterone-binding pocket are much less conserved. Therefore, the accuracy of the homology models of the progesterone-binding pocket is unclear, and the rigor of the docking studies based on the homology models is uncertain.

Moreover, on Page 16, in the 'Substrate docking and MD simulation for PbSRD5A, HsSRD5A1 and -2' section, the authors mentioned that substrates were docked into 500 conformers generated from MD simulations, making the docking results even more uncertain. Please justify the credibility of this whole approach, besides the observation that E57 and Y91 were shown to be important for binding from both computational and experimental studies.

3. On Page 14, in the section on "Initial model building of PbSRD5A", the authors did not explain why they needed initial model construction to solve the structure of PbSRD5A from their diffraction data.

4. The authors used only one sentence to describe their homology modeling of HsSRD5A1 and

HsSRD5A2: Page 8, “we built the homology models of HsSRD5A1 and -2 using the server for initial model building of PbSRD5A (Fig. 3a, b and Extended Data Fig. 6).” Please provide the details since homology modeling is an important part of this study. Particularly, the superimposition of these three structures showed a big movement of TM1, which is unusual for homology modeling. Speculate the origin of the movement from the modeling perspective.

Please also provide the name of the server for initial model building and relevant reference (e.g., Refs. 51 and 52).

5. NADPH locates in the central cavity. Discuss how NADPH enters the cavity. Is it because of the movement of TM1 or large-scale motions of the cytosolic loops?

6. On Page 7, in the ‘Coordination of NADPH’ section, the authors mentioned that ‘we hypothesized that monoolein probably occupied the steroid substrates binding pocket. For clear narration, we named TM1-4 as substrate-binding domain (SBD) and TM5-7 as NADPH binding domain (NDPBD) (Extended Data Fig. 1f)’. This statement is inaccurate. For example, Tyr32 and Arg34 interact with NADPH (see Extended Data Fig. 4), and it cannot be ruled out that some residues of TM5-7 may interact with the substrates.

Minor concerns:

1. On Page 5, in the section on “Functional characterizations of PbSRD5A”, the authors stated that the IC50 value was $1.59 \pm 0.19 \mu\text{M}$, which is different from the value ($1.58 \pm 0.19 \mu\text{M}$) given in the figure caption of Fig 1e. Please correct it.

2. In the figure caption for Fig. 3, “a, HsSRD5A1 homology model was ... represented in palegreen and lightblue. ... b, HsSRD5A2 homology model was generated on the basis of PbSRD5A structure.” Should it be “a, HsSRD5A1 homology model was ... represented in palegreen. ... b, HsSRD5A2 homology model was generated on the basis of PbSRD5A structure and represented in lightblue”?

3. On Page 11, in the Discussion section, the authors claimed that “The structural analysis and MD simulation results also supported our speculation that, in the presence of substrate, the extra space is limited to accommodate additional water molecules in the catalytic site to provide proton.” Please provide the MD simulation results and analysis in the manuscript.

4. Please provide the RMSD trajectory of the MD simulation for HsSRD5A2 to demonstrate that the system has reached equilibrium.

5. On Page 14, in the section on “Initial model building of PbSRD5A”, the authors stated that “To fuse these 9 alternative MSAs, we implemented a deep residual network with a strip pooling module to effectively capture long-range relationship of residual pairs. Such fuse architecture could alleviate the issue caused by a problematic MSA that could decrease the prediction accuracy”. Has this strip pooling module been validated? Please justify the credibility of this approach.

6. On Page 6, please briefly explain MvINS when this abbreviation was introduced for the first time. Similarly, on Page 11, please replace “... structural analysis and MD simulation ...” by “structural analysis and molecular dynamics (MD) simulation”.

7. On Page 14, please replace “(i) multiple MSA generation” by “(i) multiple multiple sequence alignment (MSA) generation” when the abbreviation MSA was first used. The references for trRosetta and AlphaFold on Page 14 were missing.

8. Please add Reference 52 for trRosetta on Page 5.

9. Fig. 3, figure caption: Remove the track changes in "Red stars (*) indicated the variants in the patients". Also, this sentence should be "Red stars (*) indicate the variants in patients". Please make the red stars in Fig. 3 and Fig. 4 more noticeable by using bold font.

10. Abstract: "...which shares 60.6% and 51.5% sequence similarities with human SRD5A1 and -2 respectively, ..." should be "... with human SRD5A1 and -2, respectively, ..."

11. Page 7: "... such as F17, S21 T24 and L25 on TM1, ..., and T109, A110 A113 and F116 on TM4". Should be "S21, T24 ... A110, A113 ..."

12. Page 17, "A 25× 25 × 25 Å³ box" should be replaced by "A 25× 25 × 25 Å³ box".

Reviewer #2 (Remarks to the Author):

This manuscript describes a structure of a steroid 5 α -reductase from Proteobacteria bacterium obtained at 2.0 angstrom resolution with NADPH bound. Based on sequence homology with human steroid 5 α -reductase (SRD5A) a homology model is built. Validation of the presumptive roles of critical residues involved in cofactor binding and steroid 5 α -reduction was conducted using site-directed mutagenesis and transfection in HEK293T cells. Evidence is provided for a conserved catalytic mechanism with steroid 5 β -reductase; and the role of disease related mutants observed in the human enzyme are elaborated. The manuscript is of potential importance since there is a deficit of structural information on the human enzyme which is targeted for the treatment of BPH with finasteride and dutasteride. In addition steroid 5 α -reductase (SRD5A2) deficiency in humans is associated with the autosomal recessive disorder of sex development associated with pseudo-hermaphroditism, lack of male pattern baldness, and an atrophied prostate gland. There is a lot to like about this manuscript but some important features require attention.

1. The authors restrict their discussion to the role of steroid 5 α -reductase to the metabolism of androgens and progestins but do not mention the importance of the enzyme in the metabolism of glucocorticoids. Please rectify. In the Main paragraph there are some incomplete thoughts. For example, the authors suggest that progesterone binds to the GABA receptor when the neurosteroid in question is the product of 5 α -dihydroprogesterone metabolism, allopregnanolone. Please correct. Similarly the authors state that androstenedione is the precursor of androgens and estrogens but do not complete the thought; are they trying to state that by conducting 5 α -reduction of these steroids the amount of androstenedione that can be converted to testosterone and estrogens will be diminished ?

2. The crystal structure reported was obtained from concentrated 5 α -reductase containing NADPH. However, the final structure likely contains NADP⁺ due to oxidation of the cofactor that will occur. This is not a trivial point since NADPH is non co-planar while NADP⁺ is aromatic and this difference will likely affect the trajectory of hydride transfer.

3. In the supplemental material the authors show the purity of their 5 α -reductase enzyme. Did the authors monitor the enzyme activity during the purification to show that they did not lose activity as part of the purification process? It would be customary to show a purification Table to show that there was an increase in specific enzyme activity with significant recovery of total units. Please provide. This is an important point to show that a large part of the enzyme was not denatured.

4. The authors show that their enzyme is only weakly inhibited by finasteride. Whereas the human enzyme displays nM affinity for this drug. This difference deserves further comment since it raises issues that the bacterial enzyme may not be such a good model of the human enzyme after all. It is

also known that finasteride is turned over by SRD5A1 to produce a high affinity enzyme generated bisubstrate analog which does not seem to be the case in their study [see, J. Amer. Chem. Soc., 1996, 118, 2359-2365].

5. If the catalytic mechanism of 5a-reductase and AKR1D1 are conserved, did the authors overlay the key residues from each structure to show the rmsd differences in the residues?

6. In docking the substrate testosterone into their structure to generate a ternary complex, how many binding poses were observed and how did they compare in energy?

7. The properties of the mutants were compared in HEK293 cells. How do the authors know that each of the mutant enzymes were equally expressed? Loss of enzyme activity could occur if some mutants were less stable. So there has to be some method used to compensate for differences in protein expression. This could have been achieved with epitope tagging with FLAG-tags.

8. The author should move Fig. 9 from the supplemental material to the body of the manuscript. This figure describes the catalytic mechanism for 5a-reductase which is a main point of the manuscript. In describing the mechanism in the discussion, the properties of the E57Q mutant seem poorly described. Surely, E57 is present in the protonated state to facilitate hydride transfer to the C3 carbonyl and this property would be lost in Q57 where an amide is present instead. Please clarify.

9. In the opinion of this reviewer the link to the COVID-19 pandemic is overstated. The paper has many attributes and this could just get a passing mention. TMPRSS2 expression is usually up regulated in late stage prostate cancer often following ADT; and expression differences may exist across ethnic groups.

10. The structure has not been deposited in the PDB which is a criteria for publication.

Minor:

1. NADPH dependent oxidoreductases do not belong to the family EC1.3.1.22. This is just the enzyme commission number where instead a family suggests some evolutionary relationship.

2. NADPH is not an electron and proton donor it donates a hydride ion.

3. 5a-androstenedione is misspelled; it should be 5a-androstanedione.

4. Please define AKR1D1 at first mention.

Reviewer #3 (Remarks to the Author):

The work is original and highly relevant for the field. The methodologies used show expertise and rigour to obtain results that only high calibre researchers would be able to. The interpretation of the data is relevant as well as its conclusions. Extensive structural modelling was used not only to allow the crystal structure solution by developing a search model for molecular replacement as well as modelling the human target based on the bacterial structure. However, those models have not been publicly published and referenced. I strongly advise that to be done to allow others in the field to have access to the modelled coordinates for scrutiny & reproducibility as well as further analysis not in scope at the time of this publication.

Research is relevant and shines light into a puzzling and important aspect of human physiology with implications in human health in a variety of forms from mental health, to growth, reproductive, wellbeing and other highly debilitating diseases. There are reports of a relation of testosterone with

regulation of the ACE2 receptor, the main target of SARS-COV-2 and related virus for cell invasion. The structural looks sound with clear presence of NADPH in the electron density and the conclusions made are reasonable. Biochemical investigations using point mutations provided further light of the structural implications on the reductase enzymatic activity enhancing our understanding of the mechanism(s) involved. The latter point allowed significant categorisation of known loss of function with mutations in the Human SRD5A2

Questions and issues for the Authors to address

Q1: Are there any drugs in the market that target SRD5A ? Would you envisage this molecule to ever be a target in the future?

Q2: A large-scale BLAST and investigation was performed to select a good candidate for studies of the HsSRD5A however no mention of what criteria were used to rank or eliminate candidates and why PbSRD5A was selected. This should be explored even if briefly. It is also not clear how many targets were selected together with PbSRD5A and why 4 particular bacterial homologs were selected.

Q3: Sequence similarity between PbSRD5A and HsSRD5A is mentioned as between 52% and 51% overall but no mention of the sequence between SBD and NDPBD domains?

Q4: Proteobacteria bacterium is a gram-negative bacterium from the same type as Escherichia, Salmonella. What was the reason to use an insect cell expression system on a protein from this organism?

Q5: Can you provide a comment on occupancy and/or B factor values of the NADPH and monoolein molecules in comparison with the surrounding atoms and residues particularly those at hydrogen or Van der Waals distances?

Q6: Can you provide any further indication (biochemical or structural) that the pocket where monoolein was modelled is the location where steroid substrates would bind? Are there any mutation work in the literature related to this ?

Q7: Have you attempted or considered obtained a complex with a inhibitor or substrate ?

Q8: Please consider depositing your homology models for HsSRD5A1 and -2 under one of the servers that accept protein structural models (e.g. <https://modelarchive.org/> or similar) and reference their unique ID in the publication. This will allow further scrutiny and reuse of those models by other researchers in the future. If you decide against this suggestion please justify?

Q9: Were there any attempts to solve the structure with sequence homolog domains before the model generation using Rosetta was attempted?

Q10: Please explain what you mean by "All diffraction data analyses have been reproduced at least three times" as states under the Reporting Summary?

Q11: Please comment why Extended Table 1 describes a resolution range for the data from 30 to 2 Å while the validation report shows 19.36 to 2.0? Was there any manual truncation of the low-resolution data and if so why?

Q12: Reporting Summary vs Extended Data table 1 vs validation report

Q13: On the reporting summary it states "For structure refinement, 10% data were selected randomly for cross validation. "

But on the Reporting Data Table 1 it states

"R_{free} was calculated with 5% of the reflections selected" (page 12, line 6-7) While on the validation report it states

2000 reflections (8.71%)

Please explain discrepancies and correct if necessary.

Q14: Please reconsider interpretation of water molecules in light of close contacts

i) A:402:HOH close contact to ARG 237

ii) A:401:HOH close contact to ASP 240

Q15: Ligands structure: There are quite a few outliers in bond length as well as large angle outliers (e.g. 133 degrees for a sp² 120 degrees expected angle is a large outlier) particularly for NADPH in the validation report. It looks no dictionary was used or the weight was too low. Can you comment if a restrain dictionary has been used for NADPH and MO? If so what weight was it used?

Other minor points

Hs in HsSRD5A is never described as "Homo sapiens". Should be defined the first time it is mentioned in the main text.

Pb in PbSRD5A is loosely defined as from Proteobacteria bacterium in the Abstract. Should be defined the first time it is mentioned in the Intro.

Page 3, second paragraph

A space is required between EC and 1.3.1.22

Page 5

Native diffraction data is collected at a certain energy, certain temperature, etc but not collected at 2.0 Angsts resolution. Unless the data was collected at a particular distance that only recorded diffraction up to 2 Angsts resolution. On that context please comment on what criteria was used for the high resolution limit of the data?

Please consider depositing your models for PbSRD5A used to perform MR on one of the servers that accept protein structural models (e.g. <https://modelarchive.org/> or similar) and reference their unique ID in the publication. This will allow further scrutiny and reuse of those models by other researchers in the future.

There is no Table 1. Maybe the reference should be Extended Data Table 1?

Describe further what "Dali search" means and reference (programs or website)

Page 6:

MVINS is mentioned for the first time without prior full description of what protein it refers too

Page 8:

Please explain and reformulate what you mean by "using the server for initial model building". Reference any website if required.

Page 13:

Was the protein-lipid reconstitution really made (v/v) or was it (w/w) or (w/v)? Please reference what protocol was used for lipid cubic phase reconstitution with reference.

Crystal size: "and reach full size within one week at 20 °C" mention the size that "full size" references to.

State that PEG 400 % is (v/v)

Page 14:

MSA is mentioned for the first time with our prior full description of the abbreviation

Page 15:

Was these data collected at one ore more beamlines? SSRL beamlines are mentioned but then only one beamline BL18U1 reference

Is there a beamline paper describing BL18U1 ? If so please reference. If not please mention in the paper or extended data the type of goniometer, beam size, an estimation of dose (or flux and exposure time) and detector used to collect the data.

Page 16:

Docking was done with AutoDock Vina. What was used for the molecular dynamics?

Page 17:

Reference the "web server H++" ?

Reference charmm36m force field?

Figure 1, page 23

c) "Time curve of reduction reaction by PbSRD5A to progesterone"

Is 5alpha-DHP being reduced to progesterone or the other way around? Please re-word to clarify what is being reduced and catalysed by what. Can you provide Km and Vmax? How do they compare with HsSRD5A?

Extended data fig 1.a, Legend

b) Crystal should be plural "crystals. Add bar with scale on (b) and (e) for assessment of crystal size.

c) What does "The initial predicted model of PbSRD5A" mean? Was this the search model for MR?

e) Colour by spectrum.? Rainbow?

f) unless defined elsewhere defined what the other colours and 3 letter codes mean (e.g TM1, ..., ECH, etc)

Extended data fig 2. Legend

PDB IDs but particularly for MaSR1 should be re-stated in the legend (actually as it is done correctly for AKR2D1 in Extended data figure 7b). Short mention of the software used and method to generated SSM should be mentioned and referenced. There are colours in the figure not mentioned in the legend. Please complete.

Extended Data Table 1

Please provide for the low resolution bin

Range

Rmerge

Rpim

CC1/2

I/Sigma

Please provide how many reflections were used for R and RFree overall but also for low and high resolution bins.

Please provide ISa for dataset.

Please make number of decimal places consistent (e.g. B factors for protein and ligand, etc)

Reviewer #4 (Remarks to the Author):

The authors report the crystal structure of a bacterial orthologue of the human steroid 5 α -reductase. The data show that the bacterial enzyme crystalizes as a seven-transmembrane structure with three transmembranes forming a putative steroid substrate binding domain and the other four composing an NADPH-cofactor binding domain. Biochemical experiments suggest the bacterial enzyme functions as a monomer to reduce the Δ 4,5 double bond of some steroids such as progesterone but not others, such as testosterone. These substrate preferences share similarities and differences with the human steroid 5 α -reductases. Similarly, the bacterial and human enzymes differ in their sensitivities to two 4-azasteroid inhibitors that are used in the clinic. A number of mutations identified in subjects with the human genetic disease steroid 5 α -reductase 2 deficiency are re-created in the bacterial enzyme and shown to have similar detrimental effects. All together, the current findings provide new insight into the structure and function of steroid 5 α -reductase, an important gene, enzyme, and pharmacological target.

Specific Comments

1. The Abstract indicates that the determined structure "unveiled the substrate recognition of SRD5A". As no steroid substrates were visualized in the structure, it might be more accurate to indicate that the structure predicted the location of a steroid substrate binding domain.
2. Similarly, the Abstract indicates in the last sentence that the deduced SRD5A structure will aid the design of therapeutic molecules. Given that hundreds if not thousands of inhibitors of SRD5A have already been identified and at least two of these are used in the clinic, it is unlikely that the atomic structure will have much impact on drug design.
3. On page 3, the first paragraph indicates that androstenedione is the precursor of androgens and estrogens, which is correct. The next sentence indicates that "These steroids could be converted to their corresponding 3-oxo-5 α steroids...". This is not a true statement as only androgens can be converted to a 3-oxo-5 α steroids; estrogens cannot be.
4. It is interesting that the bacterial enzyme is active against progesterone but not testosterone. There is some literature to suggest that over evolutionary time, progestins preceded androgens as active steroid hormones thus the bacterial enzyme substrate preference supports this literature.
5. The authors use the term "pseudovaginal perineoscrotal hypospadias" to describe the human deficiency of steroid 5 α -reductase. This term has now been replaced with "steroid 5 α -reductase 2 deficiency" as the symptoms of the disease range from simple hypospadias to the most severe form presenting with pseudovaginal perineoscrotal hypospadias.

Minor Comment

1. Was the 10x-His tag cleaved from the baculovirus-expressed SRD5A protein prior to crystallization?

Responses to the reviewers

Reviewer #1:

Steroid 5 α -reductases (SRD5 α s) are enzymes involved in catalyzing steroids with 3-oxo- Δ 4 structure, such as testosterone, androstenedione and progesterone, to generate corresponding 3-oxo-5 α steroids, which are essential for multiple physiological and pathological processes. Abnormal activities of SRD5 α s have been linked to several diseases, such as benign prostatic hyperplasia, prostatic cancer and male infertility. Recently, SRD5 α s have also been indicated as potential targets for the treatment of COVID-19. Although human steroid 5 α -reductase isozymes, HsSRD5A1 and HsSRD5A2, have been extensively investigated for decades, the structural information and molecular reaction mechanism remain poorly understood. In this paper, the authors reported the crystal structure of PbSRD5A in complex with the cofactor NADPH at high resolution (2.0 Å). PbSRD5A is a bacterial homolog of SRD5A from Proteobacteria, which shares 60.6% and 51.5% sequence similarities with HsSRD5A1 and HsSRD5A2, respectively. Homology-based structural models of HsSRD5A1 and HsSRD5A2 were built, and substrates were docked to PbSRD5A, HsSRD5A1 and HsSRD5A2, respectively. Extensive biochemical characterizations were also carried out, using in vitro reduction assay for PbSRD5A and cell-based enzymatic assay for HsSRD5A2. These structural, computational and biochemical studies were combined to understand the mechanism of NADPH mediated steroids 3-oxo- Δ 4 reduction. The presentation of the manuscript should be significantly improved to make the manuscript easier to follow. There are also many typos.

Major concerns:

1. On Page 4, in the section on “Functional characterizations of PbSRD5A”, the authors stated that “To unveil the molecular mechanisms underlying the substrate recognition and catalytic reaction of SRD5A1 and -2, BLAST searches, using HsSRD5A1 and -2 as queries, against the sequenced bacterial genomes were performed to identify a steroid 5 α -reductase suitable for structural investigation.” It seems that the authors aimed to unveil the mechanisms of substrate recognition and catalytic reaction of HsSRD5A1 and -2. However, they selected PbSRD5A, a bacterial homolog, as the target of investigation. Please explain in the main text why HsSRD5A1 or -2 were not selected as the target for investigation directly.

Reply: We thank the reviewer for the constructive suggestion to better explain the logic for homolog selection. The main reason for homolog screening is that, we have difficulties in human SRD5A1 and -2 purification due to the instability and low expression level when we initiate this project. So, we took a different approach to search a homolog, with the similar function, from lower species to get enough protein for structural studies. We also stated this directly in the revised version (Page 4, line 8-9)

2. On Page 8, in the section on “Molecular mechanism of 3-oxo- Δ 4 reduction”, the authors stated that “Considering the sequence similarities of bacterial SRD5A, HsSRD5A1 and -2, we built the homology models of HsSRD5A1 and -2 ...” and “To fully explore the substrate recognition and reduction reaction mechanism, progesterone was docked into PbSRD5A, as well as testosterone into HsSRD5A1 and -2, respectively.” The substrate selectivity suggests that different SRD5As may adopt different conformations in the substrate binding sites. Indeed, sequence alignment analysis of SRD5As across different species indicated that while the residues involved in the binding of NADP are highly conserved, the residues in the progesterone-binding pocket are much less conserved. Therefore, the accuracy of the homology models of the progesterone-binding pocket is unclear, and the rigor of the docking studies based on the homology models is uncertain.

Moreover, on Page 16, in the ‘Substrate docking and MD simulation for PbSRD5A, HsSRD5A1 and -2’ section, the authors mentioned that substrates were docked into 500 conformers generated from MD simulations, making the docking results even more uncertain. Please justify the credibility of this whole approach, besides the observation that E57 and Y91 were shown to be important for binding from both computational and experimental studies.

Reply: We appreciate the reviewer for the critical question about homology modeling and docking results. Firstly, we carefully checked the conservation of putative substrate binding pockets again, picked all residues which compose the substrate binding pockets, and highlighted the conserved residues in Table R1 as below.

Transmembrane Helix No.	Residue in human SRD5A1	Residue in human SRD5A2	Residue in PbSRD5A	Conservation
TM1	-	-	F17	Variable
	-	-	S21	
	V27	-	L25	
ICL1	S36	S31	A30	
		R94	P31	
	Y38	Y33	Y32	Identical
W56	W53	W50		
Q59	Q56	Q53		

	E60	E57	E54	
TM3	Y95	Y91	Y87	
	R98	R94	R90	
	A116	L111	T109	Variable
M119	R114	L112		
A120	G115	A113		
F123	F118	F116		
C124	C119	N117		
TM7	F221	F216	F215	Identical
	F224	F219	F218	
	F228	F223	N222	Variable

Table R1: Conservation analysis of human SRD5A1, -2 and PbSRD5A substrate binding pockets

Then we used PbSRD5A as the template to map the residues in the structure (Figure R1). It clearly showed that, the residues which are deeply buried inside pockets are highly conserved and the variable residues are gathered on periplasmic side of TM1 and TM4. Considering the orientation of NADPH and substrates, those conserved residues could recognize 3-oxo and delta-4 signature structures of the substrates and the variable residues probably will interact with the “tail” part of substrates, which is also quite different among testosterone, androstenedione and progesterone. So we believe that, our structural information, combining with the docking results, will be a good template to study not only the reaction mechanism but also substrate recognition specificity.

Indeed, we set extensive trials to get structures of all three proteins (HsSRD5A1, HsSRD5A2 and PbSRD5A) with different substrates and carried out MD simulations to identify some residues on TM1 and TM4 for substrate recognition simultaneously. However, these parts are challenging, time consuming and need massive computing resources.

Figure R1: Substrate binding pocket residues in PbSRD5A. The identical residues among human SRD5A1, -2 and PbSRD5A are highlighted in magenta and the variable residues are highlighted in yellow.

In this work we performed two different kinds of docking: (1) using crystal structure for PbSRD5A-progesterone complex; (2) using model for human SRD5A1 and -2. In both cases, flexibility of the ligands and residues were considered.

The formers follow the standard AutoDock 4 and AutoDock Vina docking procedures. Briefly, each ligand is docked with at least 250,000 poses and the calculations were repeated for 100 times. Rather than simply looking at the docking scores, poses with higher docking scores were further clustered according to their RMSDs. This gives the self-consistent poses for all ligands. Note that AutoDock 4 and AutoDock Vina yield consistent results under this analysis.

The following procedure is a more complicated ensemble docking, based on a conformation ensemble considering the thermodynamic effect obtained from all-atom MD simulations at isothermal-isobaric condition, which has shown to be effective in sampling unseen, druggable pockets for multiple targets (see Nat. Common. , 4 (2013) , p. 1407 and PLoS Negl Trop Dis. 2010 Aug 24;4(8):e803.). This is a critical step to assess the credibility of our modeling structures and predicted binding poses starting from crystal structure snapshot. In brief, all the MD simulation parameters were set up to mimic the physiological conditions of SRD5As (ER membrane, pH 7.4, and 0.15M NaCl), and the state-of-the-art force field for membrane proteins, Charm36m, was employed to describe the systems. To avoid artificial/unphysical conformational deformation of

the conserved binding pose of the NADPH inside proteins in simulations, a harmonic potential (500 kJ/mol) was applied to restraint the movement of the heavy atoms of the NADPH molecule. Moreover, given that our biochemical results have shown the catalytic activity of the NADPH/HsSRD5As to testosterone, a harmonic potential (50 kJ/mol) was applied to constraint the distance of the center-of-mass of between the nicotinamide group of NADPH and the C=O group of the testosterone predicted by the initial docking pose, ensuring the efficiency of our MD sampling on the reaction active state of the system. After a 10000 steps of energy minimization, 200 ns MD simulation was performed to fully relax the protein, membrane, and solvent step by step (Figure R2).

Figure R2: RMSD of the HsSRD5A1/NADPH and HsSRD5A2/NADPH in complexed with the testosterone along the MD simulations.

After those carefully treatments, product conformation ensemble was generated from another 50 ns of MD simulation at isothermal-isobaric ensemble, which allows us to consider thermodynamic effect on the conformation of the proteins avoiding large (high energy) conformation changes (see Nat. Commun., 4 (2013), p. 1407 and PLoS Negl Trop Dis. 2010 Aug 24; 4(8):e803.). Then, flexible docking was carried out for each structure of the conformation ensemble using AutoDock Vina. Note that exhaustiveness docking (repeated 80 times) was carried out for each conformation, and then only the best (strongest binding affinity) docking result was recorded for data analysis later on. Finally, we reported two representative binding poses for each system: a) Conf1, the one with the highest binding affinity predicted by Vina scoring function from the whole conformation ensemble (500 structures); b) Conf2, the representative of a cluster of conformations which has the highest binding affinity on average. The relationship between the representative binding poses and other (high energy) binding poses was characterized by a correlation analysis of their binding affinity and RMSD of the ligand relative to the representative conformation Conf1 (Figure R3).

Figure R3: A correlation analysis of the binding affinity and the RMSD of ligand relative to the one in the representative conformation Conf1. Here, each data point represents the average values (RMSD and/or binding affinity) of a cluster; the standard deviations of RMSD and binding affinity within each cluster are shown as error bars.

Our results show clearly that the two representative binding poses of testosterone on HsSRD5A2/HADPH complex are very similar in 3D space (upper panel of Figure R4) and their binding affinity scored by Vina are very close (-12 kcal/mol and -11.3 kcal/mol). This is also similar for the simulation of HsSRD5A1/NADPH (lower panel of Figure R4). Note that in both cases, the ligand is stabilized by a “Q-E-Y” motif of the protein. These binding poses are also very similar to the one starting from experimental conformations.

Figure R4: Two representatives (Conf1 and Conf2) of the most stable binding pose of testosterone onto NADPH/HsSRD5A1 and NADPH/HsSRD5A2 complexes.

Moreover, not only the two representatives, we found that the top 10 conformations scored by binding affinities in Vina also have similar binding poses (Figure R5), which confirms again that this interaction pattern is reliable

even the protein is relaxed. Overall, we conclude that upon binding with NADPH the pocket is conditioned for substrate binding and the subsequent reaction.

Figure R5: Superposition of the top 10 binding poses of the ligand onto (a) NADPH/HsSRD5A1 and (b) NADPH/HsSRD5A2.

3. On Page 14, in the section on “Initial model building of PbSRD5A”, the authors did not explain why they needed initial model construction to solve the structure of PbSRD5A from their diffraction data.

Reply: We appreciate the reviewer’s suggestion in better explaining the structural determination process. Using X-ray crystallography to determine the protein structure, we typically get 2D images with diffraction spots, referring to as the reciprocal lattice and representing a wave with an amplitude and a relative phase. Unfortunately, we measured the intensity of each spot but lost information about the relative phases of different diffraction. This is called “phase problem” in crystallography. Some methods like molecular replacement (MR), single-wavelength anomalous scattering (SAD) or multi-wavelength anomalous scattering (MAD) of heavy atoms in crystals are commonly used to get the phase information. Because we use lipidic cubic phase crystallization method to yield protein crystal, the crystal is growing in a gel-like lipidic environment, which is quite difficult to soak with heavy atoms. We also failed to get the phase by using seleno-methionine substitution method. Additionally, there is no structural template with reasonable sequence similarity available to try molecular replacement. So, we built the initial model from scratch for molecular replacement purpose.

4. The authors used only one sentence to describe their homology modeling of HsSRD5A1 and HsSRD5A2: Page 8, “we built the homology models of HsSRD5A1 and -2 using the server for initial model building of

PbSRD5A (Fig. 3a, b and Extended Data Fig. 6).” Please provide the details since homology modeling is an important part of this study. Particularly, the superimposition of these three structures showed a big movement of TM1, which is unusual for homology modeling. Speculate the origin of the movement from the modeling perspective.

Please also provide the name of the server for initial model building and relevant reference (e.g., Refs. 51 and 52).

Reply: We appreciate the reviewer for the critical questions about homology modeling and the movement of TM1.

For the initial PbSRD5A model building, similar to trRosetta [1], we fold the 3D model using a de novo approach from a predicted distance/orientation matrix, which is derived from a variety of multiple sequence alignments (MSAs) by a multi-branch fusion ResNet [2] (Figure R6). These MSAs are constructed from three different sequence databases, UniClust30 (UC) [3], UniRef90 (UR) [4], and NR [5] by the corresponding sequence search tools (HHblits [6], JackHmmer [7], and PSI-BLAST [8]). Given the predicted distance/orientation matrix, 300 models are constructed using PyRosetta [9] and the top 1 of them are selected according to the Rosetta energy. This model was further fed to molecular replacement (MR) to determine the protein structure.

Figure R6: A de novo approach from a predicted distance/orientation matrix, which is derived from a variety of multiple sequence alignments (MSAs) by a multi-branch fusion ResNet. Following the trRosetta methodology, we generated 300 models from the predicted distances and orientations using constrained

minimization, which is an embedded module from PyRosetta. Finally, the top 1 model with the lowest Rosetta energy was selected as the input model for molecular replacement (MR) to determine the protein structure.

We applied the same strategy for HsSRD5A1 and -2 model building and performed MD simulations. We noticed the big movement of TM1s in PbSRD5A structure and other two structural models (Extended Data Figure 7a), so we carefully checked the residue contacts among all transmembrane helices. Unlike other helices, the interactions among TM1, -4 and -7 are relatively weak, especially for the N-terminal half of TM1. This weak interaction might be the origin of TM1 flexibility. As we discussed in Point 1, TM1 probably will NOT interact with NADPH and the ring structure of steroid molecule but may provide the substrate specificity by interacting with the tail part of steroid molecules. We speculated that, TM1 may act as a gate to control the substrate entry and determine the substrate specificity. Since we are still trying to get structures of all three proteins with different substrates, it's quite difficult to interpret the conformations of TM1 in different states (apo, substrate or inhibitor bound) precisely. We also applied these structure and models in extensive MD simulation studies to identify some residues on TM1 for ligand recognition. As we mentioned above, these parts are time consuming and need massive computing resources.

Reference:

- [1] Yang, J., Anishchenko, I., Park, H., Peng, Z., Ovchinnikov, S. and Baker, D., 2020. Improved protein structure prediction using predicted interresidue orientations. *Proceedings of the National Academy of Sciences*, 117(3), pp.1496-1503.
- [2] Chen, H., Li, Y. and Su, D., 2019. Multi-modal fusion network with multi-scale multi-path and cross-modal interactions for RGB-D salient object detection. *Pattern Recognition*, 86, pp.376-385.
- [3] Mirdita, M., von den Driesch, L., Galiez, C., Martin, M.J., Soeding, J. and Steinegger, M., 2017. Uniclust databases of clustered and deeply annotated protein sequences and alignments. *Nucleic acids research*, 45(D1), pp.D170-D176.
- [4] Suzek, B.E., Wang, Y., Huang, H., McGarvey, P.B., Wu, C.H. and UniProt Consortium, 2015. UniRef clusters: a comprehensive and scalable alternative for improving sequence similarity searches. *Bioinformatics*, 31(6), pp.926-932.
- [5] NCBI Resource Coordinators, 2012. Database resources of the national center for biotechnology information. *Nucleic acids research*, 41(D1), pp.D8-D20.
- [6] Remmert, M., Biegert, A., Hauser, A. and Soeding, J., 2012. HHblits: lightning-fast iterative protein sequence searching by HMM-HMM alignment. *Nature methods*, 9(2), pp.173-175.
- [7] Eddy, S.R., 2011. Accelerated profile HMM searches. *PLoS Comput Biol*, 7(10), p.e1002195.
- [8] Altschul, S.F., Madden, T.L., Schaffer, A.A., Zhang, J., Zhang, Z., Miller, W. and Lipman, D.J., 1997. Gapped BLAST and PSI-BLAST: a new generation of protein database search programs. *Nucleic acids research*, 25(17), pp.3389-3402.
- [9] Chaudhury, S., Lyskov, S. and Gray, J.J., 2010. PyRosetta: a script-based interface for implementing molecular modeling algorithms using Rosetta. *Bioinformatics*, 26(5), pp.689-691.

5. NADPH locates in the central cavity. Discuss how NADPH enters the cavity. Is it because of the movement of TM1 or large-scale motions of the cytosolic loops?

Reply: We appreciate the reviewer for the insightful question. To our understanding, the TM1 movement will open the entrance towards the hydrophobic lipid bilayer, which is only for hydrophobic substrate entering. NADPH is well solubilized in cytosol and may enter into the cavity by the motion of cytosolic loops. Another indirect evidence is that, there are several disease related mutations located on cytosolic loops, especially at the C-terminal loop (Figure 4). Although these loops will not interact with NADPH directly, the conformations are almost identical in PbSRD5A and MaSR1 (Extended Data Figure 2), indicating their important role in maintaining the enzyme activity. However, we didn't see the conformational change of these loops during simulation, so we didn't discuss the NADPH entry in the manuscript.

6. On Page 7, in the 'Coordination of NADPH' section, the authors mentioned that 'we hypothesized that monoolein probably occupied the steroid substrates binding pocket. For clear narration, we named TM1-4 as substrate-binding domain (SBD) and TM5-7 as NADPH binding domain (NDPBD) (Extended Data Fig. 1f)'. This statement is inaccurate. For example, Tyr32 and Arg34 interact with NADPH (see Extended Data Fig. 4), and it cannot be ruled out that some residues of TM5-7 may interact with the substrates.

Reply: We thank the editor for pointing this out and corrected in the revised version (Page 7, line 23)

Minor concerns:

1. On Page 5, in the section on "Functional characterizations of PbSRD5A", the authors stated that the IC50 value was $1.59 \pm 0.19 \mu\text{M}$, which is different from the value ($1.58 \pm 0.19 \mu\text{M}$) given in the figure caption of Fig 1e. Please correct it.

Reply: Thanks for raising this point. We have corrected the figure caption of Fig. 1e in the new version. (Page 27, line 4)

2. In the figure caption for Fig. 3, "a, HsSRD5A1 homology model was ... represented in palegreen and lightblue. ... b, HsSRD5A2 homology model was generated on the basis of PbSRD5A structure." Should it be "a, HsSRD5A1 homology model was ... represented in palegreen. ... b, HsSRD5A2 homology model was generated on the basis of PbSRD5A structure and represented in lightblue"?

Reply: Thanks for raising this point. We have corrected the figure caption of Fig. 3a and b in the new version. (Page 30, line 4 and 6)

3. On Page 11, in the Discussion section, the authors claimed that “The structural analysis and MD simulation results also supported our speculation that, in the presence of substrate, the extra space is limited to accommodate additional water molecules in the catalytic site to provide proton.” Please provide the MD simulation results and analysis in the manuscript.

Reply: Thanks for raising this point. According to the reviewer’s suggestion, we provide the MD simulation result in the revised manuscript (Extended Data Figure 10, and Page 11, Line 21).

4. Please provide the RMSD trajectory of the MD simulation for HsSRD5A2 to demonstrate that the system has reached equilibrium.

Reply: Thanks for raising this point. We have provided the RMSD trajectory of the MD simulation as below.

Fig R7: The all-atom RMSD (left panel) and backbone-atoms RMSD (right panel) trajectories of the HsSRD5A2 show that the protein is fully relaxed after 100 ns of MD simulation.

5. On Page 14, in the section on “Initial model building of PbSRD5A”, the authors stated that “To fuse these 9 alternative MSAs, we implemented a deep residual network with a strip pooling module to effectively capture long-range relationship of residual pairs. Such fuse architecture could alleviate the issue caused by a problematic MSA that could decrease the prediction accuracy”. Has this strip pooling module been validated? Please justify the credibility of this approach.

Reply: Thanks for raising this point. We have validated our MSA fusion approach for protein structure prediction under CAMEO (Continuous Automated Model Evaluation) [10], a world-leading automatic structure prediction evaluation platform which includes Robetta [11] from Baker group, HHpred [12] from Soeding group, RaptorX [13] from Jinbo group, SwissModel [14] from Torsten group, etc. Specifically, since late-2019, we have registered our approach as Server83 to Server89 in CAMEO. It is worth mentioning that since June 2020,

our registered servers have been weekly, monthly, and quarterly champion teams till now. The IDDT (local Distance Difference Test) [15] scores from the best of our servers on those targets are 59.1 (from 2020-05-08 to 2020-08-01), which greatly surpassed Robetta (IDDT value is 47.7). In the near future, we shall release our approach as a web server and make it available to the public.

- [10] Haas, J., Barbato, A., Behringer, D., Studer, G., Roth, S., Bertoni, M., Mostaguir, K., Gumienny, R. and Schwede, T., 2018. Continuous Automated Model EvaluatiOn (CAMEO) complementing the critical assessment of structure prediction in CASP12. *Proteins: Structure, Function, and Bioinformatics*, 86, pp.387-398.
- [11] Kim, D.E., Chivian, D. and Baker, D., 2004. Protein structure prediction and analysis using the Robetta server. *Nucleic acids research*, 32(suppl_2), pp.W526-W531.
- [12] Soeding, J., Biegert, A. and Lupas, A.N., 2005. The HHpred interactive server for protein homology detection and structure prediction. *Nucleic acids research*, 33(suppl_2), pp.W244-W248.
- [13] Kallberg, M., Wang, H., Wang, S., Peng, J., Wang, Z., Lu, H. and Xu, J., 2012. Template-based protein structure modeling using the RaptorX web server. *Nature protocols*, 7(8), pp.1511-1522.
- [14] Waterhouse, A., Bertoni, M., Bienert, S., Studer, G., Tauriello, G., Gumienny, R., Heer, F.T., de Beer, T.A.P., Rempfer, C., Bordoli, L. and Lepore, R., 2018. SWISS-MODEL: homology modelling of protein structures and complexes. *Nucleic acids research*, 46(W1), pp.W296-W303.
- [15] Mariani, V., Biasini, M., Barbato, A. and Schwede, T., 2013. IDDT: a local superposition-free score for comparing protein structures and models using distance difference tests. *Bioinformatics*, 29(21), pp.2722-2728.

6. On Page 6, please briefly explain MvINS when this abbreviation was introduced for the first time. Similarly, on Page 11, please replace "... structural analysis and MD simulation ..." by "structural analysis and molecular dynamics (MD) simulation".

Reply: Thanks for raising this point. We have explained these abbreviations in the revised version. (Page 6, line 16-17 and Page 11, line 22)

7. On Page 14, please replace "(i) multiple MSA generation" by "(i) multiple multiple sequence alignment (MSA) generation" when the abbreviation MSA was first used. The references for trRosetta and AlphaFold on Page 14 were missing.

Reply: Thanks for raising this point. We have explained the abbreviation and added references in the revised version. (Page 14, line 18-19 and Page 15, Line 6)

8. Please add Reference 52 for trRosetta on Page 5.

Reply: Thanks for raising this point. We have added references in the revised version. (Page 5, line 12)

9. Fig. 3, figure caption: Remove the track changes in “Red stars (*) indicated the variants in the patients”. Also, this sentence should be “Red stars (*) indicate the variants in patients”. Please make the red stars in Fig. 3 and Fig. 4 more noticeable by using bold font.

Reply: Thanks for raising this point. We have removed the track changes and made the red star noticeable in the revised version. (Page 31, line 5-6)

10. Abstract: “... which shares 60.6% and 51.5% sequence similarities with human SRD5A1 and -2 respectively, ...” should be “... with human SRD5A1 and -2, respectively, ...”

Reply: Thanks for raising this point. We have corrected in the revised version. (Page 2, line 10)

11. Page 7: “... such as F17, S21 T24 and L25 on TM1, ..., and T109, A110 A113 and F116 on TM4”. Should be “S21, T24 ... A110, A113 ...”

Reply: Thanks for raising this point. We have moved the detailed residue numbers from main text to the caption of Extended Data Fig. 5b.

Reviewer #2:

This manuscript describes a structure of a steroid 5a-reductase from Proteobacteria bacterium obtained at 2.0 angstrom resolution with NADPH bound. Based on sequence homology with human steroid 5a-reductase (SRD5A) a homology model is built. Validation of the presumptive roles of critical residues involved in cofactor binding and steroid 5a-reduction was conducted using site-directed mutagenesis and transfection in HEK293T cells. Evidence is provided for a conserved catalytic mechanism with steroid 5b-reductase; and the role of disease related mutants observed in the human enzyme are elaborated. The manuscript is of potential importance since there is a deficit of structural information on the human enzyme which is targeted for the treatment of BPH with finasteride and dutasteride. In addition steroid 5a-reductase (SRD5A2) deficiency in humans is associated with the autosomal recessive disorder of sex development associated with pseudo-hermaphroditism, lack of male pattern baldness, and an atrophied prostate gland. There is a lot to like about this manuscript but some important features require attention.

1. The authors restrict their discussion to the role of steroid 5a-reductase to the metabolism of androgens and progestins but do not mention the importance of the enzyme in the metabolism of glucocorticoids. Please rectify. In the Main paragraph there are some incomplete thoughts. For example, the authors suggest that progesterone binds to the GABA receptor when the neurosteroid in question is the product of 5a-dihydroprogesterone metabolism, allopregnanolone. Please correct. Similarly the authors state that androstenedione is the precursor of androgens and estrogens but do not complete the thought; are they trying to state that by conducting 5a-reduction of these steroids the amount of androstenedione that can be converted to testosterone and estrogens will be diminished?

Reply: We thank the reviewer to make our work more precise. We have made related amendment in our manuscript. Steroids have multiple physiological and pathological functions. Here we mainly focused on prostate disease field. Testosterone, androstenedione and progesterone are the main steroid we are interested in. Cortisol, as an important steroid, is an important substrate of SRD5As, which we added related description in the main text.

2. The crystal structure reported was obtained from concentrated 5a-reductase containing NADPH. However, the final structure likely contains NADP⁺ due to oxidation of the cofactor that will occur. This is not a trivial point since NADPH is non co-planar while NADP⁺ is aromatic and this difference will likely affect the trajectory of hydride transfer.

Reply: We appreciate the reviewer for the insightful question. Due to the resolution limit, the hydrogen is invisible in the x-ray structure. So we cannot differentiate NADP⁺ and NADPH by the electron density map. For

structure determination of PbSRD5A in complex with NADPH, we added 2mM fresh-made NADPH and 5mM DTT to maintain the reduced environment during the whole purification process, and added extra 10mM NADPH before we set up the crystallization trials. The crystals grew for one week to reach the full size (15*15*120 um) in the droplets. The molar ratio of NADPH and PbSRD5A is about 10:1 in lipidic cubic phase droplets. So it has the higher chance to get the NADPH complex but not NADP⁺. We are unable to fish crystals and extract enough NADPH to determine the redox state. Meanwhile, we tried to get the substrate bound structure with NADP⁺ to see if any difference but didn't get the structure yet. Obviously, this is inconclusive at this stage until the structure with high resolution is obtained.

3. In the supplemental material the authors show the purity of their 5a-reductase enzyme. Did the authors monitor the enzyme activity during the purification to show that they did not lose activity as part of the purification process? It would be customary to show a purification Table to show that there was an increase in specific enzyme activity with significant recovery of total units. Please provide. This is an important point to show that a large part of the enzyme was not denatured.

Reply: We appreciate the reviewer for the constructive suggestion. We reserved six samples during the purification and monitored the enzyme activities. The samples we monitored include: #1 whole cell crude extract; #2 re-suspended cell membrane after 1st round of ultracentrifugation; #3 solubilized solution in 1% purification detergent DDM; #4 supernatant after 2nd round of ultracentrifugation, #5 protein eluted from Ni-column, and #6 purified protein after size-exclusion chromatography. HPLC was used to detect the 5a-DHP% converted from progesterone, using the same units of protein. Equally protein was quantified by calculating the grey level of western blot (Extended Data Figure 1b). The purification table was added in the extended materials (Extended Data Table 1) and the activities were calculated as: Calculated activity = detected activity / (Sample volumes in an assay system *grey value / Volume loaded on western blot), and normalized by sample #6 (Extended Data Figure 1b).

Your suggestion is well taken. The purification table (Extended Data Table 1) and diagram (Extended Data Table 1b) are updated in the revised manuscript.

As the results shown, PbSRD5A maintained about 20-30% of apparent activity after affinity purification. There are several possible reasons. Firstly, the lipid environment is replaced by detergent micelle and some functional important lipid molecules may be lost during purification. A recent paper also provided the evidence to demonstrate the importance of lipids in SRD5A activity (*Endocrinology*, August 2020, 161(8):1–11). However we reconstituted protein into lipidic cubic phase (LCP) for crystallization. Currently we are unable to measure the enzyme activity *in situ* because the LCP droplet is gel like. Secondly, the endogenous enzymes which may non-

specifically catalyze progesterone were removed after affinity purification. Additionally, the protein quantification by western blot is not precise enough.

4. The authors show that their enzyme is only weakly inhibited by finasteride. Whereas the human enzyme displays nM affinity for this drug. This difference deserves further comment since it raises issues that the bacterial enzyme may not be such a good model of the human enzyme after all. It is also known that finasteride is turned over by SRD5A1 to produce a high affinity enzyme generated bisubstrate analog which does not seem to be the case in their study [see, J. Amer. Chem. Soc., 1996, 118, 2359-2365].

Reply: We appreciate the reviewer for the insightful question. We noticed the finasteride is turned over by human SRD5A2 (but not SRD5A1 in most articles and reviews) to produce a high affinity bisubstrate and the recent non-peer reviewed paper published on research square also confirmed this result.

(See <https://www.ncbi.nlm.nih.gov/pmc/articles/PMC7373137/>).

The accuracy of structural model is judged majorly by the sequence homology. The PbSDR5A shares 60.6% and 51.5% sequence similarities with human SRD5A1 and -2, respectively. We carefully checked the conservation of substrate binding pockets, picked all residues which compose the substrate binding pockets, and highlighted the conserved residues in Table R1 as below.

Transmembrane Helix No.	Residue in human SRD5A1	Residue in human SRD5A2	Residue in PbSRD5A	Conservation
TM1	-	-	F17	Variable
	-	-	S21	
	V27	-	L25	
ICL1	S36	S31	A30	Identical
		R94	P31	
	Y38	Y33	Y32	
TM2	W56	W53	W50	Identical
	Q59	Q56	Q53	
	E60	E57	E54	
TM3	Y95	Y91	Y87	Identical
	R98	R94	R90	
TM4	A116	L111	T109	Variable
	M119	R114	L112	
	A120	G115	A113	
	F123	F118	F116	
	C124	C119	N117	
	F221	F216	F215	Identical

TM7	F224	F219	F218	
	F228	F223	N222	Variable

Table R1: Conservation analysis of human SRD5A1, -2 and PbSRD5A substrate binding pockets

Then we used PbSRD5A as the template to map the residues in the structure (Figure R1). It clearly showed that, the residues which are deeply buried inside pockets are highly conserved and the variable residues are gathered on periplasmic side of TM1 and TM4. Considering the orientation of NADPH and substrates, those conserved residues could recognize 3-oxo and delta-4 signature structures of the substrates and the variable residues probably will interact with the “tail” part of substrates, which are the major different among testosterone, androstenedione, progesterone and two inhibitors.

We speculated that, finasteride may not be the good inhibitor for PbSRD5A because the tail part cannot be well recognized. From the previous studies, human SRD5A1 and -2 also showed differences in inhibitor recognition. Human SRD5A2 can be inhibited by both finasteride and dutasteride but human SRD5A1 can only be strongly inhibited by dutasteride. There is NO specific inhibitor for SRD5A1 only. Without the structural information, it's impossible to figure out the recognition specificity of these SRD5As. The inhibition specificity provides the chance for inhibitor development targeting given SRD5A protein. This is why we believe that, our structural information, combining with the docking results and biochemical analysis, will help to study not only the SRD5A reaction mechanism but also inhibitor recognition specificity of SRD5As.

Figure R1: Substrate binding pocket residues in PbSRD5A. The identical residues among human SRD5A1, -2 and PbSRD5A are highlighted in magenta and the variable residues are highlighted in yellow.

Currently, we are still trying to get structures of all three proteins with varies of inhibitors. We also applied these structures and models in extensive MD simulation studies to identify some residues on TM1 and TM4 for

inhibitor recognition. However, these parts are challenging, time consuming and need massive computing resources.

5. If the catalytic mechanism of 5 α -reductase and AKR1D1 are conserved, did the authors overlay the key residues from each structure to show the rmsd differences in the residues?

Reply: We appreciate the reviewer for the constructive suggestion. PbSRD5A-progesterone complex model and AKR1D1-progesterone structure are used as templates to show the similarity (Figure R2).

Figure R2: Superposition of NADPH in PbSRD5A and AKR1D1 structures. **a**, NADPH molecules of two structures are superimposed. The NADPH and key residues of PbSRD5A are shown in green. The NADPH and key residues of AKR1D1 are shown in cyan. **b**, NADPH molecules of two structures are superimposed. The NADPH and progesterone in PbSRD5A are shown in green. The NADPH and progesterone in AKR1D1 are shown in cyan.

We superimposed NADPHs in two structures and showed progesterone molecules or key residues (Try and Glu) in panel a and b, respectively. Because AKR1D1 and PbSRD5A transfer the hydride ion from alpha or beta side of progesterone to produce the corresponding 5 α or 5 β DHP, the NADPH, progesterone and key residues in PbSRD5A are almost organized as the mirror image to that in AKR1D1. The detailed distance between substrate and key residues are labeled in Figure 3 and Extended Data Figure 7.

6. in docking the substrate testosterone into their structure to generate a ternary complex, how many binding poses were observed and how did they compare in energy?

Reply: We appreciate the reviewer for the insightful question. In this work we performed two different kinds of docking: (1) using crystal structure for PbSRD5A- progesterone complex; (2) using model for human SRD5A1 and -2. In both cases, flexibility of the ligands and residues were considered.

The formers follow the standard AutoDock 4 and AutoDock Vina docking procedures. Briefly, each ligand is docked with at least 250,000 poses and the calculations were repeated for 100 times. Rather than simply looking at the docking scores, poses with higher docking scores were further clustered according to their RMSDs. This gives the self-consistent poses for all ligands. Note that AutoDock 4 and AutoDock Vina yield consistent results under this analysis.

The following procedure is a more complicated ensemble docking, based on a conformation ensemble considering the thermodynamic effect obtained from all-atom MD simulations at isothermal-isobaric condition, which has shown to be effective in sampling unseen, druggable pockets for multiple targets (see *Nat. Commun.*, 4 (2013), p. 1407 and *PLoS Negl Trop Dis.* 2010 Aug 24;4(8):e803.). This is a critical step to assess the credibility of our modeling structures and predicted binding poses starting from crystal structure snapshot. In brief, all the MD simulation parameters were set up to mimic the physiological conditions of SRD5As (ER membrane, pH 7.4, and 0.15M NaCl), and the state-of-the-art force field for membrane proteins, Charm36m, was employed to describe the systems. To avoid artificial/unphysical conformational deformation of the conserved binding pose of the NADPH inside proteins in simulations, a harmonic potential (500 kJ/mol) was applied to restraint the movement of the heavy atoms of the NADPH molecule. Moreover, given that our biochemical results have shown the catalytic activity of the NADPH/HsSRD5As to testosterone, a harmonic potential (50 kJ/mol) was applied to constraint the distance of the center-of-mass of between the nicotinamide group of NADPH and the C=O group of the testosterone predicted by the initial docking pose, ensuring the efficiency of our MD sampling on the reaction active state of the system. After a 10000 steps of energy minimization, 200 ns MD simulation was performed to fully relax the protein, membrane, and solvent step by step (Figure R2).

Figure R2: RMSD of the HsSRD5A1/NADPH and HsSRD5A2/NADPH in complexed with the testosterone along the MD simulations.

After those carefully treatments, product conformation ensemble was generated from another 50 ns of MD simulation at isothermal-isobaric ensemble, which allows us to consider thermodynamic effect on the conformation of the proteins avoiding large (high energy) conformation changes (see *Nat. Commun.*, 4 (2013),

p. 1407 and PLoS Negl Trop Dis. 2010 Aug 24; 4(8):e803.). Then, flexible docking was carried out for each structure of the conformation ensemble using AutoDock Vina. Note that exhaustiveness docking (repeated 80 times) was carried out for each conformation, and then only the best (strongest binding affinity) docking result was recorded for data analysis later on. Finally, we reported two representative binding poses for each system: a) Conf1, the one with the highest binding affinity predicted by Vina scoring function from the whole conformation ensemble (500 structures); b) Conf2, the representative of a cluster of conformations which has the highest binding affinity on average. The relationship between the representative binding poses and other (high energy) binding poses was characterized by a correlation analysis of their binding affinity and RMSD of the ligand relative to the representative conformation Conf1 (Figure R3).

Figure R3: A correlation analysis of the binding affinity and the RMSD of ligand relative to the one in the representative conformation Conf1. Here, each data point represents the average values (RMSD and/or binding affinity) of a cluster; the standard deviations of RMSD and binding affinity within each cluster are shown as error bars.

Our results show clearly that the two representative binding poses of testosterone on HsSRD5A2/HADPH complex are very similar in 3D space (upper panel of Figure R4) and their binding affinity scored by Vina are very close (-12 kcal/mol and -11.3 kcal/mol). This is also similar for the simulation of HsSRD5A1/NADPH (lower panel of Figure R4). Note that in both cases, the ligand is stabilized by a “Q-E-Y” motif of the protein. These binding poses are also very similar to the one starting from experimental conformations.

Figure R4: Two representatives (Conf1 and Conf2) of the most stable binding pose of testosterone onto NADPH/HsSRD5A1 and NADPH/HsSRD5A2 complexes.

Moreover, not only the two representatives, we found that the top 10 conformations scored by binding affinities in Vina also have similar binding poses (Figure R5), which confirms again that this interaction pattern is reliable even the protein is relaxed. Overall, we conclude that upon binding with NADPH the pocket is conditioned for substrate binding and the subsequent reaction.

Figure R5: Superposition of the top 10 binding poses of the ligand onto (a) NADPH/HsSRD5A1 and (b) NADPH/HsSRD5A2.

7. The properties of the mutants were compared in HEK293 cells. How do the authors know that each of the mutant enzymes were equally expressed? Loss of enzyme activity could occur if some mutants were less stable. So there has to be some method used to compensate for differences in protein expression. This could have been achieved with epitope tagging with FLAG-tags.

Reply: We appreciate the reviewer for the constructive suggestion. We measured the expression level by western blot using anti-human SRD5A2 antibody and updated in Extended Data Figure 8. (Page 9, Line 6)

8. The author should move Fig. 9 from the supplemental material to the body of the manuscript. This figure describes the catalytic mechanism for 5a-reductase which is a main point of the manuscript. In describing the mechanism in the discussion, the properties of the E57Q mutant seem poorly described. Surely, E57 is present in the protonated state to facilitate hydride transfer to the C3 carbonyl and this property would be lost in Q57 where an amide is present instead, Please clarify.

Reply: We appreciate the reviewer for the constructive suggestion and made changes in the revised version. (Figure 4; Page 11, line 18-21)

9. In the opinion of this reviewer the link to the COVID-19 pandemic is overstated. The paper has many attributes and this could just get a passing mention. TMPRSS2 expression is usually up regulated in late stage prostate cancer often following ADT; and expression differences may exist across ethnic groups.

Reply: We appreciate the reviewer for the constructive suggestion and removed the COVID-19 part in the revised version.

10. The structure has not been deposited in the PDB which is a criteria for publication.

Reply: We appreciate the reviewer for the constructive suggestion and deposited the structure in PDB (PDB code: 7C83)

Minor:

1. NADPH dependent oxidoreductases do not belong to the family EC1.3.1.22. This is just the enzyme commission number where instead a family suggests some evolutionary relationship.

Reply: Thanks for raising this point. We have deleted this description in the revised version. (Page 3, line 13-14)

2. NADPH is not an electron and proton donor it donates a hydride ion.

Reply: Thanks for raising this point. We have corrected in the revised version. (Page 3, line 17)

3. 5a-androstenedione is misspelled; it should be 5a-androstanedione.

Reply: Thanks for raising this point. We have corrected all this misspelled word in the revised version.

4. Please define AKRID1 at first mention.

Reply: Thanks for raising this point. We have defined AKR1D1 as steroid 5 β -reductase, Aldo-keto reductase family 1 member D1 in the revised version. (Page 8, line 19)

Reviewer #3 (Remarks to the Author):

The work is original and highly relevant for the field. The methodologies used show expertise and rigour to obtain results that only high calibre researchers would be able to. The interpretation of the data is relevant as well as its conclusions. Extensive structural modelling was used not only to allow the crystal structure solution by developing a search model for molecular replacement as well as modelling the human target based on the bacterial structure. However, those models have not been publicly published and referenced. I strongly advise that to be done to allow others in the field to have access to the modelled coordinates for scrutiny & reproducibility as well as further analysis not in scope at the time of this publication.

Research is relevant and shines light into a puzzling and important aspect of human physiology with implications in human health in a variety of forms from mental health, to growth, reproductive, wellbeing and other highly debilitating diseases. There are reports of a relation of testosterone with regulation of the ACE2 receptor, the main target of SARS-COV-2 and related virus for cell invasion. The structural looks sound with clear presence of NADPH in the electron density and the conclusions made are reasonable. Biochemical investigations using point mutations provided further light of the structural implications on the reductase enzymatic activity enhancing our understanding of the mechanism(s) involved. The latter point allowed significant categorisation of known loss of function with mutations in the Human SRD5A2.

Questions and issues for the Authors to address

Q1: Are there any drugs in the market that target SRD5A? Would you envisage this molecule to ever be a target in the future?

Reply: We appreciate the reviewer for the constructive suggestion. As far as we know, there are at least 2 drugs, finasteride and dutasteride, in the market targeting SRD5As. The indications of finasteride covers benign prostatic hyperplasia and alopecia and the indication of dutasteride is benign prostatic hyperplasia. That's why we used these two benchmark compounds to inhibit SRD5As in our work. It's worth mentioning that, dutasteride targets both SRD5A1 and -2 but finasteride specifically targets SRD5A2. We majorly focus on the molecular mechanism of SRD5As induced prostate cancer occurrence and progression. Activity of SRD5A1 but not -2 is dramatically increased in the prostate cancer patients for unknown reason. However, there is NO specific SRD5A1 inhibitor available in the market. Additionally, the current SRD5A inhibitors can be classified to steroid like and non-steroid like molecules. Both finasteride and dutasteride are steroidal medicine. Long term treatment of finasteride or dutasteride reduces the overall incidence of prostate cancer but results in more

aggressive prostate cancer. The molecular mechanism behind this outcome is still unclear. So the development of non-steroidal inhibitor may provide an alternative choice. We believe that, our structural information, combining with computational studies and biochemical analysis, will shed on the light of the inhibitor rational design. So our future work will focus on the inhibitor-SRD5A complex structure determination, biochemical analysis, as well as novel inhibitor design.

Q2: A large-scale BLAST and investigation was performed to select a good candidate for studies of the HsSRD5A however no mention of what criteria were used to rank or eliminate candidates and why PbSRD5A was selected. This should be explored even if briefly. It is also not clear how many targets were selected together with PbSRD5A and why 4 particular bacterial homologs were selected.

Reply: We appreciate the reviewer for the constructive suggestion. We briefly describe the criteria for homolog selection in the revised manuscript (Page 4, line 17-21). The criteria to rank the candidates are based on the sequence similarity, host species, expression level, profile on size-exclusion chromatography and the quality of crystals. Indeed, we blasted over 100 species, cloned 10 candidates with highest sequence similarity to human SRD5A2, selected 4 with highest expression level, and set up one round of crystallization trials. The PbSRD5A protein could yield the biggest, best diffracted crystals in about 2 days and the crystal reached to the full size (15*15*120 μm) in one week. We simply named the other 3 candidates #1, #2 and #3 here. #1 cannot yield crystal, #2 cannot catalyze 3-oxo- Δ^4 -steroids, and #3 is partly aggregated after concentration. Additionally, these 3 candidates exhibited totally different inhibition properties to finasteride and dutasteride. Although we didn't report the biochemical analysis in the manuscript, these 3 candidates may act as good candidates in the following studies to explore the substrate and inhibitor specificities.

Q3: Sequence similarity between PbSRD5A and HsSRD5A is mentioned as between 52% and 51% overall but no mention of the sequence between SBD and NDPBD domains?

Reply: We appreciate the reviewer for the constructive suggestion. The overall sequence similarity of PbSRD5A and HsSRD5A1 is 60.6%. The SBDs (TM1-4) and NDPBDs (TM5-7) of PbSRD5A and HsSRD5A1 share 50.3% and 72.3% similarity, respectively. We also took suggestion from another reviewer not to define these two domains because we didn't get the substrate-protein complex and cannot rule out the contribution of TM5-7 in substrate binding.

Q4: Proteobacteria bacterium is a gram-negative bacterium from the same type as Escherichia, Salmonella. What was the reason to use an insect cell expression system on a protein from this organism?

Reply: We appreciate the reviewer for the interesting question. We tried *E.coli* expression system first and got small amount of proteins. However, the protein profile on size-exclusion is poor and the N-terminal tag cannot be removed. Considering its high sequence similarity with human SRD5As, we tested insect cell system and got the functional protein. Besides, a recent paper also provided the evidence to show the *importance* of lipids in SRD5A activity (*Endocrinology*, August 2020, 161(8):1–11). They used *E.coli* system to get human SRD5A protein but have to reconstitute it to liposome to recover enzyme activity.

Q5: Can you provide a comment on occupancy and/or B factor values of the NADPH and monoolein molecules in comparison with the surrounding atoms and residues particularly those at hydrogen or Van der Waals distances?

Reply: We appreciate the reviewer for the insightful question. We checked the B factors of ligand key atoms and the surrounding residues. The detailed information is shown in the table below. The occupancy of the ligands and the surrounding atoms are 1.

NADPH	B factor	Surrounding atoms	B factor	Interactions
O1N	16.20	Tyr32 OH	18.12	Hydrogen Bond
N7A	17.50	Arg34 NH2	43.27	
O3D	21.90	Arg90 NH2	21.63	
O3B	25.61	Arg99 O	37.05	
O3D	21.90	Asn159 ND2	18.72	
ND2	16.27	Asp163 OD1	21.74	
O2X	19.11	Arg170 NH2	28.15	
O2B	22.55	Arg170 NH1	38.15	
O1X	21.70	Tyr177 OH	26.80	
N6A	22.78	Glu178 O	35.72	
O1A	22.46	Asn192 ND2	16.27	
O2D	16.09	Glu196 OE2	19.48	
O2A	17.92	His230 NE2	19.70	
N6A	22.78	Tyr234 OH	27.48	
Average	21.40			
Monoolein	B factor	Surrounding atoms	B factor	Interactions
O25	24.05	Gln53 OE1	24.56	Hydrogen Bond
O23	55.49	Trp50 NE1	23.99	Hydrophobic interactions
		Phe17	43.43	
		Leu25	24.36	
		Ser21	30.48	
		Tyr32	18.05	
		Ala113	24.87	
		Phe116	26.53	
		Phe218	31.23	
Average	47.94			

Q6: Can you provide any further indication (biochemical or structural) that the pocket where monoolein was modelled is the location where steroid substrates would bind? Are there any mutation work in the literature related to this?

Reply: We appreciate the reviewer for the insightful question. Indeed we tried very hard to get the substrate or inhibitor bound structure but didn't get any promising results. One of the possible reason is that, the monoolein concentration is much higher than the solubility of substrates and inhibitors. We added inhibitors and substrates in the purification buffers, monoolein and crystallization buffers but cannot replace monoolein in the structures. The recent non-peer reviewed paper published on research square supported the pocket is the real binding site for substrates and inhibitors (see <https://www.ncbi.nlm.nih.gov/pmc/articles/PMC7373137/>). Because human SRD5A2 will form a bisubstrate complex with finasteride but not the case in PbSRD5A, we also tried to co-crystallize dutasteride and other inhibitors with those SRD5As. Moreover, the disease mutations we mentioned in the manuscript can be classified into 3 categories. Q53, E54, and Y89 in PbSRD5A2 located in the monoolein binding site but won't interact with NADPH. The good size-exclusion profiles of these 3 mutants (Q56A, E54L and Y89F) indicated that, the overall structures are maintained. This is also an indirect evidence to confirm the substrate binding site.

Q7: Have you attempted or considered obtained a complex with an inhibitor or substrate?

Reply: We appreciate the reviewer for the constructive suggestion. We tried very hard and are still attempting to get complex structures with inhibitors and substrates using HsSRD5As and other homologs.

Q8: Please consider depositing your homology models for HsSRD5A1 and -2 under one of the servers that accept protein structural models (e.g. <https://modelarchive.org/> or similar) and reference their unique ID in the publication. This will allow further scrutiny and reuse of those models by other researchers in the future. If you decide against this suggestion please justify?

Reply: We appreciate the reviewer for the constructive suggestion. We have deposited the initial model of PbSRD5A and two human SRD5A models in *modelarchive*. The model IDs are ma-xs1jw (PbSRD5A), ma-bfecj (human SRD5A1), and ma-ib3wq (human SRD5A2).

Q9: Were there any attempts to solve the structure with sequence homolog domains before the model generation using Rosetta was attempted?

Reply: We appreciate the reviewer for the insightful question. We tried MR using MaSR1 TM5-10 but failed. We also tried Se-met method but cannot get enough proteins for crystallization. Soaking with heavy metal is quite challenging because LCP droplets will change to lamella phase, which have strong diffraction spots under X-ray, in a few seconds.

Q10: Please explain what you mean by "All diffraction data analyses have been reproduced at least three times" as states under the Reporting Summary?

Reply: Thanks for raising this question. The diffraction data we used are processed by two authors (Xiao, Q and Ren, R.) independently. Before submission to PDB, Xiao re-processed data again to confirm the accuracy.

Q11: Please comment why Extended Table 1 describes a resolution range for the data from 30 to 2 Å while the validation report shows 19.36 to 2.0? Was there any manual truncation of the low-resolution data and if so why?

Reply: Thanks for raising this question. In merge and scale process by aimless, we set the resolution from 30 Å to 2 Å. However, the spots with a resolution lower than 20Å were blocked by beamstop in BL18U1 if the distance of the detector is less than 400mm. There isn't manual truncation of low-resolution data.

Q12: Reporting Summary vs Extended Data table 1 vs validation report

Reply: Thanks for raising this question. We have submitted the updated reporting summary, validation report and revised Extended Data tables.

Q13: On the reporting summary it states

"For structure refinement, 10% data were selected randomly for cross validation." But on the Reporting Data Table 1 it states Rfree was calculated with 5% of the reflections selected" (page 12, line 6-7) While on the validation report it states 2000 reflections (8.71%). Please explain discrepancies and correct if necessary.

Reply: Thanks for raising this question. We corrected the number in revised manuscript from 5% to 10%. We re-examined the parameter for the refine process. In default, 10% data were selected to calculate the Rfree in Phenix. However, Rfree was calculated with 8.71% of the reflections selected in practice. The following figure lists the diffraction data for R_{work} and R_{free} during the refine process.

REMARK	3	FIT TO DATA USED IN REFINEMENT (IN BINS).								
REMARK	3	BIN	RESOLUTION	RANGE	COMPL.	NWORK	NFREE	RWORK	RFREE	CCWOR
REMARK	3	1	19.3656	- 4.7977	0.99	1597	153	0.2189	0.2633	0.90
REMARK	3	2	4.7977	- 3.8176	1.00	1543	147	0.2002	0.2495	0.89
REMARK	3	3	3.8176	- 3.3379	1.00	1515	145	0.1696	0.2061	0.93
REMARK	3	4	3.3379	- 3.0339	1.00	1511	143	0.1791	0.1872	0.92
REMARK	3	5	3.0339	- 2.8172	1.00	1508	144	0.1643	0.1930	0.93
REMARK	3	6	2.8172	- 2.6515	1.00	1484	142	0.1719	0.2139	0.93
REMARK	3	7	2.6515	- 2.5190	1.00	1499	142	0.1779	0.2311	0.92
REMARK	3	8	2.5190	- 2.4096	1.00	1487	142	0.1810	0.2332	0.92
REMARK	3	9	2.4096	- 2.3170	0.99	1477	141	0.1926	0.2219	0.91
REMARK	3	10	2.3170	- 2.2371	1.00	1463	139	0.1868	0.2354	0.91
REMARK	3	11	2.2371	- 2.1673	1.00	1483	142	0.1994	0.2659	0.90
REMARK	3	12	2.1673	- 2.1054	0.99	1503	144	0.2167	0.2518	0.89
REMARK	3	13	2.1054	- 2.0500	0.99	1451	138	0.2393	0.2839	0.88
REMARK	3	14	2.0500	- 2.0001	0.99	1450	138	0.2877	0.3249	0.82

Q14: Please reconsider interpretation of water molecules in light of close contacts

i) A:402:HOH close contact to ARG 237

ii) A:401:HOH close contact to ASP 240

Reply: Thanks for raising this question. We re-examined two pairs of molecules. The distance between 402:HOH and NH2 atom on ARG237 is 2.25 Å. The distance between 401:HOH and O atom on ASP240 is 2.2Å. We corrected the positions of the two water molecules in Coot. The minimum distance between 402:HOH and ARG237 is 2.62 Å. The distance between 401:HOH and O atom on ASP240 is 2.43 Å. We will update the structure in PDB.

Q15: Ligands structure: There are quite a few outliers in bond length as well as large angle outliers (e.g. 133 degrees for a sp2 120 degrees expected angle is a large outlier) particularly for NADPH in the validation report. It looks no dictionary was used or the weight was too low. Can you comment if a restrain dictionary has been used for NADPH and MO? If so what weight was it used?

Reply: Thanks for raising this question. Ligand restraints was generated by the PHENIX.eLBOW and used during the refinement. Targets and weighting were not selected in the Phenix.refine. The default value of Refinement Weight is 60 in Coot.

Other minor points

Hs in HsSRD5A is never described as "Homo sapiens". Should be defined the first time it is mentioned in the main text. Pb in PbSRD5A is loosely defined as from Proteobacteria bacterium in the Abstract. Should be defined the first time it is mentioned in the Intro.

Reply: Thanks for raising this question. We have defined Hs and Pb in the main text. (Page 4, Line 11 and 15)

Page 3, second paragraph A space is required between EC and 1.3.1.22

Reply: Thanks for raising this question. Another reviewer pointed that, “*NADPH dependent oxidoreductases do not belong to the family EC1.3.1.22. This is just the enzyme commission number where instead a family suggests some evolutionary relationship.*” So we made this change in the revised version. (Page 3, Line 13)

Page 5: Native diffraction data is collected at a certain energy, certain temperature, etc but not collected at 2.0 Angsts resolution. Unless the data was collected at a particular distance that only recorded diffraction up to 2 Angsts resolution. On that context please comment on what criteria was used for the high resolution limit of the data?

Reply: Thanks for raising this question. The criteria is that, signal to noise ratio is greater than 2, the CC1/2 is greater than 0.5, the integrity is greater than 90%, and R_{merge} is less than 1. We corrected the description in the revised version. (Page 5, Line 11)

Please consider depositing your models for PbSRD5A used to perform MR on one of the servers that accept protein structural models (e.g. <https://modelarchive.org/> or similar) and reference their unique ID in the publication. This will allow further scrutiny and reuse of those models by other researchers in the future.

Reply: Thanks for raising this questions. We have deposited the initial model of PbSRD5A and two human SRD5A models in *modelarchive* and referenced in the manuscript. The model IDs are ma-xs1jw (PbSRD5A), ma-bfecj (human SRD5A1), and ma-ib3wq (human SRD5A2).

There is no Table 1. Maybe the reference should be Extended Data Table 1?

Reply: Thanks for raising this questions. We've corrected in the revised version. (Extended Data Table 2)

Describe further what “Dali search” means and reference (programs or website)

Reply: Thanks for raising this questions. We've added the website in the manuscript. (Page 5, Line 23)

Page 6: MvINS is mentioned for the first time without prior full description of what protein it refers too

Reply: Thanks for raising this point. We have explained these abbreviations in the revised version. (Page 6, Line 16-17)

Page 8: Please explain and reformulate what you mean by “using the server for initial model building”.

Reference any website if required.

Reply: Thanks for raising this point. Here we listed the performance of tFold, the unpublished server we used, on the CAMEO (Continuous Automated Model Evaluation) website: https://www.cameo3d.org/modeling/1-week/difficulty/all/?to_date=2020-10-03. For the last 6 months, our server ranked 1st, compared with benchmark servers such as Robetta and SWISS-MODEL. We will publish and share the server soon.

Page 13: Was the protein-lipid reconstitution really made (v/v) or was it (w/w) or (w/v)? Please reference what protocol was used for lipid cubic phase reconstitution with reference.

Reply: Thanks for raising this point. We have corrected the reconstitution of protein-lipid ratio of 2:3 (w/w) and listed the reference in the manuscript. (Page 14, Line 11)

Crystal size: “and reach full size within one week at 20 °C” mention the size that “full size” references to. State that PEG 400 % is (v/v)

Reply: Thanks for raising this point. We have added the crystal size and stated the unit of PEG400 solution. (Page 14, Line 14 and 15)

Page 14:MSA is mentioned for the first time with our prior full description of the abbreviation

Reply: Thanks for raising this point. We have explained the abbreviation (Page 14, Line 18-19)

Page 15:Was these data collected at one or more beamlines? SSRL beamlines are mentioned but then only one beamline BL18U1 reference. Is there a beamline paper describing BL18U1 ? If so please reference. If not

please mention in the paper or extended data the type of goniometer, beam size, an estimation of dose (or flux and exposure time) and detector used to collect the data.

Reply: Thanks for raising this point. We collected data sets from BL18U1 in SSRF only. Currently there is not a beamline paper describing BL18U1. So we added the beamline parameters in the main text. (Page 15, line 16-18)

Page 16: Docking was done with AutoDock Vina. What was used for the molecular dynamics?

Reply: Thanks for raising this question. All the MD simulations were conducted by Gromacs 2019.04. We have referenced in the revised manuscript. (Page 17, Line 19)

Page 17: Reference the "web server H++"? Reference charmm36m force field?

Reply: Thanks for raising this point. We have referenced in the manuscript. (Page 17, Line 23 and Page 18, Line 1)

Figure 1, page 23: c) "Time curve of reduction reaction by PbSRD5A to progesterone" Is 5alpha-DHP being reduced to progesterone or the other way around? Please re-word to clarify what is being reduced and catalyzed by what. Can you provide Km and Vmax? How do they compare with HsSRD5A?

Reply: Thanks for raising this point. The reaction is one way from progesterone to 5alpha-DHP. The Km of purified PbSRD5A is 12.1 μM and Vmax is 0.4 $\mu\text{mol}/\text{min}^{-1}\text{mg}^{-1}$ enzyme for progesterone with NADPH as cofactor at pH 7.0 (Figure R1). The reported Km of HsSRD5A2 is 0.9 μM and Vmax is 1.9 $\text{nmol}/\text{min}^{-1}\text{mg}^{-1}$ enzyme for testosterone with NADPH as cofactor at pH 6.0. (See Makridakis *et al.*, *Pharmacogenetics* 10, 407-413 (2000)). Because of the difference of substrates and detection method, it's hard to compare the kinetics of two enzymes.

Figure R1: The Double reciprocal Plots of purified PbSRD5A kinetics.

Extended data fig 1.a, Legend

b) Crystal should be plural "crystals. Add bar with scale on (b) and (e) for assessment of crystal size.

c) What does "The initial predicted model of PbSRD5A " mean? Was this the search model for MR?

e) Colour by spectrum.? Rainbow?

f) unless defined elsewhere defined what the other colours and 3 letter codes mean (e.g TM1, ..., ECH, etc)

Reply: Thanks for raising these points. We have corrected the typos, added the scale bars and defined abbreviations in the manuscript. (Supplementary materials Page 3)

Extended data fig 2. Legend

PDB IDs but particularly for MaSR1 should be re-stated in the legend (actually as it is done correctly for AKRID1 in Extended data figure 7b). Short mention of the software used and method to generated SSM should be mentioned and referenced. There are colors in the figure not mentioned in the legend. Please complete.

Reply: Thanks for raising these points. We have completed colors, re-stated PDB code of MaSR1 and briefly described the method we used to generate SSM. (Supplementary materials Page 4)

Extended Data Table 1

Please provide for the low resolution bin: Range, Rmerge, Rpim, CCI/2, I/Sigma,

Please provide how many reflections were used for R and RFree overall but also for low and high resolution bins.

Please provide ISa for dataset.

Please make number of decimal places consistent (e.g. B factors for protein and ligand, etc)

Reply: Thanks for raising these points. We have updated the Extended Data table 2.

Reviewer #4 (Remarks to the Author):

The authors report the crystal structure of a bacterial orthologue of the human steroid 5 α -reductase. The data show that the bacterial enzyme crystalizes as a seven-transmembrane structure with three transmembranes forming a putative steroid substrate binding domain and the other four composing an NADPH-cofactor binding domain. Biochemical experiments suggest the bacterial enzyme functions as a monomer to reduce the Δ 4,5 double bond of some steroids such as progesterone but not others, such as testosterone. These substrate preferences share similarities and differences with the human steroid 5 α -reductases. Similarly, the bacterial and human enzymes differ in their sensitivities to two 4-azasteroid inhibitors that are used in the clinic. A number of mutations identified in subjects with the human genetic disease steroid 5 α -reductase 2 deficiency are re-created in the bacterial enzyme and shown to have similar detrimental effects. All together, the current findings provide new insight into the structure and function of steroid 5 α -reductase, an important gene, enzyme, and pharmacological target.

Specific Comments

1. The Abstract indicates that the determined structure “unveiled the substrate recognition of SRD5A”. As no steroid substrates were visualized in the structure, it might be more accurate to indicate that the structure predicted the location of a steroid substrate binding domain.

Reply: We appreciate the reviewer for the constructive suggestion. We have corrected in the revised version. (Page 2, line16)

2. Similarly, the Abstract indicates in the last sentence that the deduced SRD5A structure will aid the design of therapeutic molecules. Given that hundreds if not thousands of inhibitors of SRD5A have already been identified and at least two of these are used in the clinic, it is unlikely that the atomic structure will have much impact on drug design.

Reply: We appreciate the reviewer for the insightful question. As far as we know, there are 2 drugs, finasteride and dutasteride, in the market targeting SRD5As. The indications of finasteride covers benign prostatic hyperplasia and male androgenetic alopecia and the indication of dutasteride is benign prostatic hyperplasia. It's worth mentioning that, dutasteride targets both SRD5A1 and -2 but finasteride specifically targets SRD5A2. However there is NO specific SRD5A1 inhibitor available in the market. We majorly focus on the molecular

mechanism of SRD5As induced prostate disease. Activity of SRD5A1 but not -2 is dramatically increased in the prostate cancer patients for unknown reason. Based on our structural, biochemical and MD simulation information, the tail part of steroid like inhibitors are essential in improving the recognition specificity and potency. The variable residues of human SRD5A1 and -2 gathered in TM1 and TM4 are responsible for interacting with the tail of inhibitors. We believe that, our structural information, combining with computational studies and biochemical analysis, will shed on the light of the inhibitor rational design targeting specific isoforms. Additionally, both finasteride and dutasteride are steroidal inhibitors and long term treatment of these drugs leads to an increase incidence of aggressive prostate cancer. The molecular mechanism behind this outcome is still unclear. One possible reason is the steroidal metabolism of these inhibitors by varies of enzymes *in vivo*. So the non-steroidal inhibitor development may provide an alternative way for cancer prevention. Our future work will focus on the inhibitor-SRD5A complex structure determination, biochemical analysis, as well as novel inhibitor design.

3. On page 3, the first paragraph indicates that androstenedione is the precursor of androgens and estrogens, which is correct. The next sentence indicates that "These steroids could be converted to their corresponding 3-oxo-5 α steroids...". This is not a true statement as only androgens can be converted to a 3-oxo-5 α steroids; estrogens cannot be.

Reply: We appreciate the reviewer for the constructive suggestion. Combining with other reviewer's suggestions, we restated the first paragraph in the revised manuscript. (Page 3, line 4-11)

4. It is interesting that the bacterial enzyme is active against progesterone but not testosterone. There is some literature to suggest that over evolutionary time, progestins preceded androgens as active steroid hormones thus the bacterial enzyme substrate preference supports this literature.

Reply: We appreciate the reviewer for the interesting discussion. Here we briefly describe the criteria for homolog selection. The criteria to rank the candidates are based on the sequence similarity, host species, expression level, profile on size-exclusion chromatography and the quality of crystals. Indeed, we blasted over 100 species, cloned 10 candidates with highest sequence similarity to human SRD5A2, selected 4 with highest expression level, and set up one round of crystallization trials. We also conducted biochemical analysis for all 4 candidates. We simply named other 3 candidates #1, #2 and #3 here. #1 can efficiently catalyze progesterone, testosterone and androstenedione but didn't yield crystal. #2 cannot catalyze 3-oxo- Δ 4-steroids at all, although shared over 50% sequence similarity. #3 can catalyzed progesterone, testosterone and androstenedione but the

protein partly aggregated after concentration. The inhibition patterns are different as well. So these candidates may provide good chance in the following studies to explore the substrate and inhibitor specificities.

5. The authors use the term “pseudovaginal perineoscrotal hypospadias” to describe the human deficiency of steroid 5 α -reductase. This term has now been replaced with “steroid 5 α -reductase 2 deficiency” as the symptoms of the disease range from simple hypospadias to the most severe form presenting with pseudovaginal perineoscrotal hypospadias.

Reply: Thanks for raising this point. We have corrected in the manuscript. (Page 9, Line 9 and Page 10, Line 12)

Minor Comment

1. Was the 10x-His tag cleaved from the baculovirus-expressed SRD5A protein prior to crystallization?

Reply: Thanks for raising this point. The tag was removed before the protein was injected to size-exclusion chromatography. The cutting site is “DEVDA” and could be cleaved between D and A by drICR. We have specified this in the manuscript. (Page 13, Line23)

REVIEWER COMMENTS

Reviewer #1 (Remarks to the Author):

The authors have addressed my questions. The revised version is significantly improved.

Reviewer #2 (Remarks to the Author):

This manuscript is much improved, unfortunately a number of issues remain. These include scientific and editing. With regards to the latter there is poor use of the English Language especially in the new sections. A careful read of the revision raises the following points that remain. Nonetheless the manuscript still represents a major contribution.

Scientific Issues:

1. The authors were asked to provide a purification table for PbSRD5A. However, it is impossible to determine whether their purified enzyme has retained the expected activity. Fig. 1 (extended data) shows loss of total activity at different purification stages. This is to be expected since it is a recovery issue. However, one would like to see that throughout the purification there is an increase in specific activity $\mu\text{moles}/\text{min}/\text{mg}$. If the specific activity is not increasing the authors have not enriched for activity indicating that a large portion of their isolated protein may be inactive. The purification table should show specific activity at each stage.
2. If the PbSRD5A reduces progesterone but not testosterone, what is the explanation? The major interest in human steroid 5 α -reductase is the reduction of testosterone to 5 α -dihydrotestosterone. As this reaction is not observed this diminishes the value of drawing parallels with the human enzyme. This point needs to be addressed.
3. In the abstract they indicate that the work would lead to more specific inhibitors of steroid 5 α -reductase. This issue has been solved with finasteride and dutasteride. If the authors want a specific inhibitor only for SRD5A1 they should make the case as to why this is needed. The opinion of this reviewer is that the strength of the article is in the structural information on an important steroidogenic enzyme that has been lacking and that this information can be used to assign the properties of disease-related mutants. It is this point that should be stressed.
4. Why do the authors claim that 5 α -androstenedione regulates prostate function? (p3. Line 7)
5. The authors were asked to correct the statement that 5 α -dihydroprogesterone is a neurosteroid when in fact the steroid is allopregnanolone (3 α -hydroxy-5 α -pregnane-20-one). Please correct. (p3 Line 8)
6. "Metabolism of steroidal medicine, such as abiraterone and galeterone" should read: "the metabolism of P450c17 inhibitors used to treat castrate resistant prostate cancer, e.g. abiraterone and galeterone" (p3. Line 11).
7. Potent reduction activity is jargon. Do the authors mean catalytic efficiency ($k_{\text{cat}}/K_{\text{m}}$)? (p5. Line 1)
8. "Traceable activity to androstenedione" is meaningless. The substrate should be 4-androstene-3,17-dione and "traceable" should have a number or is it below the limit of detection?. If so what is the limit of detection? (p5 Line 2)
9. "finasteride showed only mild inhibition only at high concentration" (p5 line 6). Should read finasteride was a weak inhibitor yielding an IC_{50} value = x.

10. "It suggested the evolutionary conservation of NADPH binding". What is the basis of this statement?

11. The authors draw parallels with the catalytic mechanism for AKR1D1. In the AKR superfamily including AKR1D1, the catalytic Tyr is the principal proton donor and Glu plays a facilitatory role in polarizing the carbonyl group. However, the authors suggest that E57 is the proton donor. Unless the authors have evidence to the contrary, they should modify the mechanism proposed.

Editorial

1. Please correct androstenedione to 4-androstene-3,17-dione throughout.
2. "and was majorly coordinated" poor English (p7. Line 5)
2. That took input multiple sequence alignments not clear

Trevor M. Penning, PhD

Reviewer #3 (Remarks to the Author):

In general I am happy with the modifications made on the manuscript as well as with the answers provided. A few minor points remain to be addressed.

Page 9, Line 174

The submission of the homology models to databases is significant and appreciated as it will allow further future analysis for all continuing studies on this puzzling enzymes. The explanation on "user the server for initial model building" is relevant and I understand that the work on CAMEO3D is not published however no change in the manuscript text was made to at least point to the URL provided on the answer. I feel that with it this does not adequately describe how the homology models were generated.

Page 16, line 329

Appreciate spelling out MSA but there is now twice the word "multiple" in the text

Page 17, line 352

replace "*" with "x" on 8×10^{11} and units with ph/s

Page 18, line 396

The reference for GROMACS is still missing

Reviewer #4 (Remarks to the Author):

The authors have addressed the issues raised in the initial critique in a satisfactory manner. This is an important paper that will be widely read.

Responses to the reviewers

Reviewer #2:

This manuscript is much improved, unfortunately a number of issues remain. These include scientific and editing. With regards to the latter there is poor use of the English Language especially in the new sections. A careful read of the revision raises the following points that remain. Nonetheless the manuscript still represents a major contribution.

Scientific Issues:

1. The authors were asked to provide a purification table for PbSRD5A. However, it is impossible to determine whether their purified enzyme has retained the expected activity. Fig. 1 (extended data) shows loss of total activity at different purification stages. This is to be expected since it is a recovery issue. However, one would like to see that throughout the purification there is an increase in specific activity umoles/min/mg. If the specific activity is not increasing the authors have not enriched for activity indicating that a large portion of their isolated protein may be inactive. The purification table should show specific activity at each stage.

Reply: We appreciate the reviewer for the insightful discussion. We provided the total activity, specific activity and calculated activity values (normalized) in the updated extended table 1. We calculated the total activity values as “Total product /Reaction time”, specific activity values as “Total product / (Total protein × Reaction time)” and calculated activity values (normalized) as “Detected activity / (Sample volumes in an assay system × grey value / Volume loaded on western blot)”, and normalized by sample #6.. The specific activity values indicated that active PbSRD5A were enriched throughout the purification. The calculated activity values (normalized) indicated that the PbSRD5A maintained activity during purification. Because samples 1-3 are homogenized membrane, we are unable to measure the total protein values using UV spectrometer. Thus, the specific activity values are not applicable for samples 1-3.

2. If the PbSRD5A reduces progesterone but not testosterone, what is the explanation? The major interest in human steroid 5 α -reductase is the reduction of testosterone to 5 α -dihydrotestosterone. As this reaction is not observed this diminishes the value of drawing parallels with the human enzyme. This point needs to be addressed.

Reply: We appreciate the reviewer for the insightful discussion. First of all, we briefly describe the criteria for homolog selection. The criteria to rank the candidates are based on the sequence similarity, host species, expression level, profile on size-exclusion chromatography and the quality of crystals. Indeed, we blasted over 100 species, cloned 10 candidates with highest sequence similarity to human SRD5A2, selected 4 with highest expression level, and set up one round of crystallization trials. We also conducted biochemical analysis for all 4 candidates. We simply named other 3 candidates #1, #2 and #3 here. #1 can efficiently catalyze progesterone, testosterone and androstenedione but didn't yield crystal. #2 cannot catalyze 3-oxo- Δ 4-steroids at all, although shared over 50% sequence similarity. #3 can catalyzed progesterone, testosterone and androstenedione but the protein partly aggregated after concentration. The inhibition patterns are different as well.

We have mentioned in the manuscript that the variable residues on TM1 and TM4 may be the main determinants for enzyme specificity. In the following work, a high throughput MD simulation will be performed, using 6 protein structural templates, 4 substrates and 2 inhibitors, to figure out key residues involving in substrate recognition. Extensive biochemical analysis and mutagenesis studies are needed although this a time and resource consuming process.

Additionally, some literatures suggested that over evolutionary time, progestin preceded androgens as active steroid hormones thus the bacterial enzyme substrate preference supports this literature.

In general, we believe that the bacterial structure provides information of great use to characterize human SRD5As.

3. In the abstract they indicate that the work would lead to more specific inhibitors of steroid 5 α -reductase. This issue has been solved with finasteride and dutasteride. If the authors want a specific inhibitor only for SRD5A1 they should make the case as to why this is needed. The opinion of this reviewer is that the strength of the article is in the structural information on an important steroidogenic enzyme that has been lacking and that this information can be used to assign the properties of disease-related mutants. It is this point that should be stressed.

Reply: We thank the reviewer for this suggestion and we now strength the point that the results would be used to assign the properties of disease-related mutants (Page 2, Line 16-18). For the statement of inhibitor development, there are 2 drugs, finasteride and dutasteride, in the market targeting SRD5As. The indications of finasteride covers benign prostatic hyperplasia and male androgenetic alopecia and the indication of dutasteride is benign prostatic hyperplasia. It's worth mentioning that, dutasteride targets both SRD5A1 and -2 but finasteride specifically targets SRD5A2. However there is NO specific SRD5A1 inhibitor available in the market. We majorly focus on the molecular mechanism of SRD5As induced prostate disease. Activity of

SRD5A1 but not -2 is dramatically increased in the prostate cancer patients for unknown reason. Based on our structural, biochemical and MD simulation information, the tail part of steroid like inhibitors are essential in improving the recognition specificity and potency. The variable residues of human SRD5A1 and -2 gathered in TM1 and TM4 are responsible for interacting with the tail of inhibitors. We believe that, our structural information, combining with computational studies and biochemical analysis, will shed on the light of the inhibitor rational design targeting specific isoforms. Additionally, both finasteride and dutasteride are potent SRD5A inhibitors used in clinic. However, the long term treatment of finasteride or dutasteride would increase the incidence of aggressive prostate cancer [1, 2]. Novel (non-steroidal) inhibitors might be helpful for prostate cancer management. Our future work will focus on the inhibitor-SRD5A complex structure determination, biochemical analysis, as well as novel inhibitor design.

[1] G.L. Andriole, D.G. Bostwick, O.W. Brawley, L.G. Gomella, M. Marberger, F. Montorsi, C.A. Pettaway, T.L. Tammela, C. Teloken, D.J. Tindall, M.C. Somerville, T.H. Wilson, I.L. Fowler, R.S. Rittmaster, R.S. Group, Effect of dutasteride on the risk of prostate cancer, *The New England journal of medicine*, 362 (2010) 1192-1202.

[2] I.M. Thompson, P.J. Goodman, C.M. Tangen, M.S. Lucia, G.J. Miller, L.G. Ford, M.M. Lieber, R.D. Cespedes, J.N. Atkins, S.M. Lippman, S.M. Carlin, A. Ryan, C.M. Szczepanek, J.J. Crowley, C.A. Coltman, Jr., The influence of finasteride on the development of prostate cancer, *The New England journal of medicine*, 349 (2003) 215-224.

4. Why do the authors claim that 5 α -androstenedione regulates prostate function? (p3. Line 7)

Reply: We thank the reviewer for this comment. An alternative pathway has been unveiled to find that DHT synthesis bypasses testosterone [3]. Androstenedione is converted to 5 α -androstenedione, but not testosterone, for DHT synthesis. This pathway might protect androgens from degradation by UGT2B15/17, to facilitate androgen accumulation. This pathway was found in castration resistant prostate cancer cells at first. Later, in our lab, biopsy samples from benign patients (no prostate cancer cells at all) could also generate DHT through 5 α -androstenedione, indicating that the function of 5 α -androstenedione is not limited in prostate cancer. Since these results have not been published, we revised the manuscript and emphasized the function of 5 α -androstenedione in prostate cancer only.

[3] K.H. Chang, R. Li, M. Papari-Zareei, L. Watumull, Y.D. Zhao, R.J. Auchus, N. Sharifi, Dihydrotestosterone synthesis bypasses testosterone to drive castration-resistant prostate cancer, *Proceedings of the National Academy of Sciences of the United States of America*, 108 (2011) 13728-13733.

5. The authors were asked to correct the statement that 5 α -dihydroprogesterone is a neurosteroid when in fact the steroid is allopregnanolone (3 α -hydroxy-5 α -pregnane-20-one). Please correct. (p3 Line 8)

Reply: Thanks for raising this point. We have corrected this statement. (Page 3, Line 7-9)

6. “Metabolism of steroidal medicine, such as abiraterone and galeterone” should read: “the metabolism of P450c17 inhibitors used to treat castrate resistant prostate cancer, e.g. abiraterone and galeterone” (p3. Line 11).

Reply: Thanks for raising this point. We have corrected this statement. (Page 3, Line 11-13)

7. Potent reduction activity is jargon. Do the authors mean catalytic efficiency (kcat/Km)? (p5. Line 1)

Reply: Thanks for raising this point. We have corrected this statement. (Page 5, Line 2-4) We didn't include the kinetic parameters in the MS. The preliminary result showed that, Km of purified PbSRD5A is 12.1 μ M and Vmax is 0.4 μ mol/min⁻¹mg⁻¹ enzyme for progesterone with NADPH as cofactor at pH 7.0 (Figure R1).

Figure R1: The Double reciprocal Plots of purified PbSRD5A kinetics.

8. “Traceable activity to androstanedione” is meaningless. The substrate should be 4-andostene-3,17-dione and “traceable” should have a number or is it below the limit of detection?. If so what is the limit of detection? (p5 Line 2)

Reply: Thanks for raising this point. We have corrected this statement. (Page 5, Line 2-4) We provided PbSRD5A enzyme activities to T, AD and P in Fig 1b. As figure shown, only 5% of 4-andostene-3,17-dione, but 80% of progesterone, could be catalyzed under the same condition. No 5 α -dihydrotestosterone is detected.

9. “Finasteride showed only mild inhibition only at high concentration” (p5 line 6). Should read finasteride was a weak inhibitor yielding an IC50 value = x.

Reply: Thanks for raising this point. We have corrected this statement (Page 5, Line 7). Due to the solubility limit, we are unable to provide the precise IC50 value of finasteride. Based on the previous data, the estimated IC50 of finasteride is ten times less than dutasteride at least.

10. “It suggested the evolutionary conservation of NADPH binding”. What is the basis of this statement?

Reply: Thanks for raising the question. We stated this based on the sequence alignment of SRD5A proteins of over 100 species. We mapped the highly conserved residues and found that, these residues are gathering together to coordinate NADPH or composing the putative substrate binding pocket (Sup Fig 9). We also analyzed the delta (14)-sterol reductase structure and found the residues for NADPH binding are also conserved with PbSRD5A.

11. The authors draw parallels with the catalytic mechanism for AKR1D1. In the AKR superfamily including AKR1D1, the catalytic Tyr is the principal proton donor and Glu plays a facilitory role in polarizing the carbonyl group. However, the authors suggest that E57 is the proton donor. Unless the authors have evidence to the contrary, they should modify the mechanism proposed.

Reply: Thanks for raising this insightful discussion. We didn't test the AKR1D1 mutants to figure out which residue is the proton donor. However, we proposed that E57 is the proton donor based on our mutagenesis analysis. E57 should be protonated to facilitate polarization of the carbonyl group, which is well stated in the previous literatures. If E57 only performs facilitory role, the patient mutation E57Q should maintain enzyme activity. Based on our biochemical results (Fig 3f), we proposed that the proton may come from E57. Besides, we excluded the possibility of water molecule as proton donor by analyzing the accessibility using MD simulation (Sup Fig 10).

Editorial

1. Please correct androstenedione to 4-androstene-3,17-dione throughout.

Reply: Thanks for raising this point. We have corrected throughout.

2. “and was majorly coordinated” poor English (p7. Line 5)

Reply: Thanks for raising this point. We have corrected this statement. (Page 7, Line 5)

3. That took input multiple sequence alignments not clear

Reply: Thanks for raising this point. The 3D model building started from an alignment method called “multiple MSA”. MSA means multiple sequence alignment. We restated in the new version. (Page 14, Line 21)

Reviewer #3 (Remarks to the Author):

In general I am happy with the modifications made on the manuscript as well as with the answers provided. A few minor points remain to be addressed.

(Page 9, Line 174) The submission of the homology models to databases is significant and appreciated as it will allow further future analysis for all continuing studies on this puzzling enzymes. The explanation on "user the server for initial model building" is relevant and I understand that the work on CAMEO3D is not published however no change in the manuscript text was made to at least point to the URL provided on the answer. I feel that with it this does not adequately describe how the homology models were generated.

Reply: Thanks for raising this point. We revised the model building part in method and provided the URLs of three models in Modelarchive (Page 15, Line 15). We also informed the Modelarchive team to release the details. Currently, the models are available to public.

(Page 16, line 329) Appreciate spelling out MSA but there is now twice the word "multiple" in the text

Reply: Thanks for raising this point. The 3D model building started from an alignment method called “multiple MSA”. MSA means multiple sequence alignment. We used upper case letters of “Multiple Sequence Alignment” to better present this in the new version. (Page 14, Line 19)

(Page 17, line 352) replace "" with "x" on 8×10^{11} and units with ph/s*

Reply: Thanks for raising this point. We have corrected. (Page 16, Line 10)

(Page 18, line 396) The reference for GROMACS is still missing

Reply: Thanks for raising this point. The reference is inserted in the revised MS. (Page 18, Line 8)

REVIEWER COMMENTS

Reviewer #2 (Remarks to the Author):

The authors have responded to my critique well except as it relates to catalytic mechanism. They still propose that E57 is the principal proton donor. However, their own mutagenesis data show that both Y91F and Y91D eliminate enzyme activity while residual activity remains in the E57Q mutant which is more consistent with Y91 acting as the general acid. Furthermore, examination of the crystal structure of AKR1D1 shows a dual function for the corresponding glutamic acid residue (it also allowed steroid substrates to penetrate the active site more deeply so that hydride transfer could occur to C5). How can the authors be so confident that Y91 is not the general acid and that E57 facilitates enolization of the carbonyl and allows penetration of the steroid into the PbSDR5A active site? Unless I am missing something the authors do not have compelling data to distinguish between these mechanisms.

Reviewer #3 (Remarks to the Author):

I am satisfied with the answers provided by the authors.

Responses to the reviewers

Reviewer #2:

The authors have responded to my critique well except as it relates to catalytic mechanism. They still propose that E57 is the principal proton donor. However, their own mutagenesis data show that both Y91F and Y91D eliminate enzyme activity while residual activity remains in the E57Q mutant which is more consistent with Y91 acting as the general acid. Furthermore, examination of the crystal structure of AKR1D1 shows a dual function for the corresponding glutamic acid residue (it also allowed steroid substrates to penetrate the active site more deeply so that hydride transfer could occur to C5). How can the authors be so confident that Y91 is not the general acid and that E57 facilitates enolization of the carbonyl and allows penetration of the steroid into the PbSDR5A active site? Unless I am missing something the authors do not have compelling data to distinguish between these mechanisms.

Reply: We thank the reviewer's insightful discussion. We missed to emphasize the importance of Y91 in human SRH5A2 to facilitate enolization of the carbonyl and made a significant change in the new MS.

Here I would like to discuss the proton donor to complete the reduction reaction based on the structural comparison with AKR1D1 and SRD5A2. Figure R1 showed the proposed reaction mechanism of SRD5A2 (Herbert G. Bull et al., *JACS*, 1996).

Figure R1. Proposed mechanism of human SRD5A2.

In general, the proton (represented by B-H in figure R1) may come from Tyr, Glu or H₂O in the pocket. We rule out the possibility of H₂O to provide the proton in the presence of substrate by structural analysis and MD simulation. Shown in figure R2, NADPH transfer the hydride ion from beta-face of testosterone in AKR1D1 and alpha-face in SRD5As to achieve stereo-specificity. Except for hydride, another proton should be added at C4 of testosterone. The Glu and Tyr residues are pointed to the alpha-face of testosterone in AKR1D1 and beta-face in SRD5As. We measured the distances of Glu and Tyr to C4 of testosterone in AKR1D1 and SRD5A2 and showed in Table R1. In SRD5A2, hydrogen atom of carboxyl group in E57 side chain (3.5 Å) is closer to C4 of testosterone than hydroxyl group of Y91 (4.9 Å). To the contrast, hydrogen atom in hydroxyl group of Y58 (3.5 Å) is closer to C4 of testosterone than carboxyl group in E120 (4.6 Å). So we proposed the proton may be transferred from E57 in SRD5A2 to C4 of testosterone.

Figure R2: Superposition of NADPH in PbSRD5A and AKR1D1 structures. **a**, NADPH molecules of two structures are superimposed. The NADPH and key residues of PbSRD5A are shown in green. The NADPH and key residues of AKR1D1 are shown in cyan. **b**, NADPH molecules of two structures are superimposed. The NADPH and progesterone in PbSRD5A are shown in green. The NADPH and progesterone in AKR1D1 are shown in cyan.

AKR1D1		SRD5A2	
O1 (Glu120)	6.6 Å	O1 (Glu57)	3.6 Å
O2 (Glu120)	4.6 Å	O2 (Glu57)	3.5 Å
O (Tyr58)	3.5 Å	O (Tyr91)	4.9 Å

Table R1. Distances of oxygen in glutamate and tyrosine side chains to C4 of Testosterone.

REVIEWERS' COMMENTS

Reviewer #2 (Remarks to the Author):

The authors have now dealt with my criticism and the mechanism of catalysis proposed is now reasonable.